# Variance Reduced Smoothed Functional REINFORCE Policy Gradient Algorithms

**Shalabh Bhatnagar**  *shalabh@iisc.ac.in*
*Department of Computer Science and Automation,*
*Indian Institute of Science,*
*Bengaluru 560012, India*

**Deepak Ramachandra**  *deepakh@iisc.ac.in*
*Department of Computer Science and Automation,*
*Indian Institute of Science,*
*Bengaluru 560012, India*

**Reviewed on OpenReview:** *https://openreview.net/forum?id=yagxqSJbiY*

## Abstract

We revisit the REINFORCE policy gradient algorithm from the literature that works with reward (or cost) returns obtained over episodes or trajectories. We propose a major enhancement to the basic algorithm where we estimate the policy gradient using a smoothed functional (random perturbation) gradient estimator obtained from direct function measurements. To handle the issue of high variance that is typical of REINFORCE, we propose two independent enhancements to the basic scheme: (i) use the sign of the increment instead of the original (full) increment that results in smoother convergence and (ii) use clipped gradient estimates as proposed in the Proximal Policy Optimization (PPO) based scheme. We prove the asymptotic convergence of all algorithms and show the results of several experiments on various MuJoCo locomotion tasks wherein we compare the performance of our algorithms with the recently proposed ARS algorithms in the literature as well as other well known algorithms namely A2C, PPO and TRPO. Our algorithms are seen to be competitive against all algorithms and in fact show the best results on a majority of experiments.

**Key Words:** smoothed functional REINFORCE policy gradient algorithms, stochastic shortest path Markov decision processes, signed updates, objective function clipping.

## 1 Introduction

Policy gradient methods (Sutton et al., 1999; Sutton & Barto, 2018) are a popular class of approaches in reinforcement learning (Bertsekas, 2019; Meyn, 2022). Parameter-dependent randomized policies are normally used in these approaches and one updates the policy parameter along a gradient search direction. The policy gradient theorem (Sutton et al., 1999; Marbach & Tsitsiklis, 2001; Cao, 2007), which is a fundamental result in these approaches, relies on an interchange of the gradient and expectation operators and in such cases turns out to be the expectation of the gradient of noisy performance functions, much like the perturbation analysis-based methods studied previously for simulation optimization (Ho & Cao, 1991; Chong & Ramadge, 1993).

The REINFORCE algorithm (Williams, 1992; Sutton & Barto, 2018) is a noisy gradient scheme for which the updates of the policy parameter are obtained once after the full return on an episode has been found. Actor-critic algorithms (Sutton & Barto, 2018; Konda & Borkar, 1999; Konda & Tsitsiklis, 2003; Bhatnagar et al., 2009; 2007) have been presented in the literature as alternatives to the REINFORCE algorithm as they perform incremental parameter updates at every instant.

We revisit the REINFORCE algorithm in this paper and present modified zeroth order algorithms as alternatives to it. While REINFORCE type algorithms are trajectory-based methods, they are single timescale algorithms unlike actor-critic schemes that involve two timescales. REINFORCE is a first-order gradient search algorithm that is based on direct gradient measurements. First-order gradient search algorithms in data-driven optimization work well in situations where the gradient of the expected (noisy) performance objective can be shown equal to the expectation of the gradient of the (noisy) performance function. REINFORCE is one such scheme but because it is a trajectory-based method, suffers usually from high variance.

In many real-life situations, particularly those involving infinite state spaces, interchanging the gradient and expectation operators (mentioned above) may not be justified without additional requirements related to the smoothness of the (noisy) performance objective as well as other restrictions on the system dynamics (Chong & Ramadge, 1994; 1993). Zeroth-order algorithms, on the other hand, do not suffer from this problem as the expectation of the noisy sample performance functions is directly obtained from the effects of the stochastic approximation scheme and the gradient (in the limit) is then estimated from the converged expected values, thereby invalidating the need for an interchange between the gradient and expectation operators. This motivates us to propose an efficient zeroth-order algorithm as an alternative to REINFORCE. Our algorithms are devised for episodic tasks, also referred to as the stochastic shortest path setting, and perform parameter updates upon termination of episodes, that is when the goal or terminal states are reached (Cao, 2007; Marbach & Tsitsiklis, 2001). We may however mention that our gradient estimation procedure would work even for continuing task settings such as those involving the infinite horizon discounted reward criterion. A policy gradient procedure for this setting is described in Paternain et al. (2022), see Algorithms 1-2 there. The gradient estimator there involves the computation of a complex matrix-valued function. Zeroth-order methods can potentially help in this process as one would not need to compute such quantities.

Our first algorithm is based on a single function measurement or simulation at a perturbed parameter value where the perturbations are obtained using independent Gaussian random variates. The problem, however, is that it suffers from a large bias in the gradient estimator. We show analytically the reason for the large bias here. Subsequently, we also present the two-function measurement variant of this scheme which we show, in a result, has lower bias. Our algorithms rely on a diminishing sensitivity parameter sequence $\{\delta_n\}$ that appears in the denominator of an increment term in our algorithms. This can result in high variance at least in the initial iterates. To tackle this problem, we introduce the signed analogs of these algorithms where we only consider the sign of the increment terms (the ones that multiply the learning rates in the updates).

Subsequently, we also incorporate variants that use gradient clipping as with the proximal policy optimization (PPO) algorithm (Schulman et al., 2017). A similar scheme as our first (single-measurement) algorithm is briefly presented in Bhatnagar (2023) that, however, does not present any analysis of convergence or experiments. Our paper not only provides a detailed analysis and experiments with the one-measurement scheme, but also analyzes several other related algorithms both for their convergence and empirical performance. We do not analyze, however, the finite-time convergence of our algorithms. We refer the reader to Yuan et al. (2022) for a sample complexity analysis in the discounted reward setting of REINFORCE with regular PG estimates. Gradient estimation in our algorithm is performed using the smoothed functional (SF) technique for gradient estimation (Rubinstein, 1981; Bhatnagar & Borkar, 2003; Bhatnagar, 2007; Bhatnagar et al., 2013). The basic problem in this setting is the following: Given an objective function $J : \mathcal{R}^d \to \mathcal{R}$ such that $J(\theta) = E_\xi[h(\theta, \xi)]$, where $\theta \in \mathcal{R}^d$ is the parameter to be tuned and $\xi$ is the noise element, the goal is to find $\theta^* \in \mathcal{R}^d$ such that $J(\theta^*) = \max_{\theta \in \mathcal{R}^d} J(\theta)$. Since the objective function $J(\cdot)$ can be highly nonlinear, one often settles for a lesser goal – that of finding a local instead of a global maximum.

In Salimans et al. (2017), evolutionary strategy (ES), also sometimes referred to as basic random search (BRS) based zeroth order gradient estimation algorithms involving one and two measurement smoothed functional estimators have been proposed as alternatives to the REINFORCE algorithm. One of the measurements in the two-measurement estimator is the running parameter making the estimator one-sided (instead of the two-sided estimator that we use). During each run of the algorithm, in the ES or BRS procedure, a certain number ($k$) of gradient estimates is obtained by randomly sampling the search directions and an average over the $k$ gradient estimates is then used in the procedure. In Mania et al. (2018a;b), the augmented random

search (ARS) procedures are proposed as modifications to the ES scheme where the best $b$ out of the $k$ directions are used and the average over these samples is further divided by the standard deviation of the $2b$ returns. These algorithms are seen to show good results. We also implement the ARS algorithms in our work. Further, in prior work, asymptotic convergence analyses of ES or ARS had not been provided. We thus also provide the first asymptotic convergence analysis of ES/ARS algorithms.

In Malik et al. (2020), smoothed functional algorithms (both one and two simulation) are applied for policy optimization in the setting of a linear quadratic regulator (LQR) problem on linear policies and non-asymptotic regret bounds are obtained. The cost function in this setting is seen to satisfy nice properties such as the Polyak-Lojasiewicz (PL) condition. Unlike Malik et al. (2020), we do not restrict ourselves to linear policies or to linear state evolution dynamics as our state process follows general nonlinear dynamics and our cost function can be highly nonlinear and non-convex with multiple local optima. We provide asymptotic analyses of all algorithms including ARS in this setting. Most of our analysis is based on just two assumptions, namely Assumptions 1 and 2. In fact, we prove all the basic requirements such as the parameterized value function being differentiable with a Lipschitz continuous gradient (Lemma 4). This is unlike papers on ES/ARS that make much stronger assumptions but do not show whether such assumptions are valid in the settings they consider.

Random search methods such as simultaneous perturbation stochastic approximation (SPSA) (Spall, 1992; 1997; Bhatnagar, 2005), smoothed functional (SF) (Katkovnik & Kulchitsky, 1972; Bhatnagar & Borkar, 2003; Bhatnagar, 2007) or random directions stochastic approximation (RDSA) (Kushner & Clark, 1978; Prashanth et al., 2017) have the advantage that they typically require only one or two system simulations to estimate the objective function gradient regardless of the parameter dimension $d$. Textbook treatments of random search approaches for stochastic optimization are available in (Spall, 2005; Bhatnagar et al., 2013; Prashanth & Bhatnagar, 2025).

In addition to proving the asymptotic convergence, we empirically study the performance of our algorithms along with their clipped and signed variants with the ARS algorithms on four different MuJoCo locomotion tasks, namely, Swimmer, Hopper, HalfCheetah and Walker2d, respectively. It has been observed in the past (Mania et al., 2018a;b) that ARS algorithms show significantly better performance on most tasks than other model-free algorithms such as PPO, TRPO and A2C. We thus focus mainly on empirical comparisons of our algorithms with ARS and we also show such comparisons with these other algorithms on some of the tasks.

## 1.1 Our Contributions

1. We present zeroth order versions of the REINFORCE policy gradient algorithm that are based on smoothed functional gradient estimators (SFR) and require one or two function measurements at each update. While SF gradient estimates have been proposed in the past, they have only so far been studied in the context of the ARS/ES algorithms.

2. Our basic scheme is similar to ARS except that ARS requires multiple simulations for a given parameter update as opposed to just one or two simulations for the algorithms that we present.

3. For improved accuracy and lower variance, we propose variants of our algorithm with clipped gradient estimates as well as signed updates.

4. We prove the asymptotic convergence of all algorithms including ARS as well as those with signed updates and clipped gradient estimators. Most of our analysis is based on just two assumptions, namely Assumptions 1 and 2. In fact, we prove all the basic requirements such as the parameterized value function being differentiable with a Lipschitz continuous gradient (Lemma 4). This is unlike papers on ES/ARS that make much stronger assumptions but do not show whether such assumptions are valid in the settings they consider.

5. We show empirical results on four different MuJoCo environments as well as on a gridworld environment with varying grid sizes. Our experiments demonstrate that our algorithms with the clipped and signed updates perform better than ARS on more than half of the settings. Our algorithm SFR-2 shows uniformly better performance than A2C, PPO and TRPO on all the gridworld settings. On the MuJoCo environments, SFR-2 shows the best performance on two of the four MuJoCo

environments (when ARS is also considered) and it is better than A2C, TRPO and PPO on three of these environments.

## 1.2 Paper Organization

The rest of the paper is organized as follows. Section 2 presents the Markov decision process (MDP) framework for the episodic setting and recalls the policy gradient theorem from Sutton & Barto (2018). Section 3 presents the basic one-simulation SFR algorithm (SFR-1). Section 4 then presents the two-simulation SFR algorithm (SFR-2) as well as its signed and clipped counterparts. We also prove a couple of lemmas here that show that SFR-2 has better bias guarantees than SFR-1 and the signed and clipped variants have better variance guarantees than the original algorithm. We also present here the two evolutionary strategy (ES) algorithms from the literature and briefly describe the ARS algorithms. Section 5 provides the main results in the convergence analysis of all algorithms. Proofs of these results are given in Appendix A. Section 6 provides detailed numerical results and Section 7 provides the concluding remarks. Appendix A has all the proofs of the various convergence results, Appendix B contains details of the ARS algorithm and qualitative comparisons with SFR, and Appendix C contains details of the setting parameters for the various numerical experiments.

## 2 The MDP Framework

By a Markov decision process (MDP), we mean a controlled stochastic process $\{X_n\}$ whose evolution is governed by an associated control-valued sequence $\{Z_n\}$. It is assumed that $X_n, n \geq 0$ takes values in a set $S$ called the state-space. Let $A(s)$ be the set of feasible actions in state $s \in S$ and $A \overset{\triangle}{=} \cup_{s \in S} A(s)$ denote the set of all actions. When the state is say $s$ and a feasible action $a$ is chosen, the next state seen is $s'$ with a probability $p(s'|s,a) \overset{\triangle}{=} P(X_{n+1} = s' \mid X_n = s, Z_n = a)$, $\forall n$. Such a process satisfies the controlled Markov property, i.e., $P(X_{n+1} = s' \mid X_n, Z_n, \ldots, X_0, Z_0) = p(s' \mid X_n, Z_n)$ a.s., $\forall n \geq 0$.

By an admissible policy or simply a policy, we mean a sequence of functions $\pi = \{\mu_0, \mu_1, \mu_2, \ldots\}$, with $\mu_k : S \to A$, $k \geq 0$, such that $\mu_k(s) \in A(s)$, $\forall s \in S$. When following policy $\pi$, a decision maker selects action $\mu_k(s)$ at instant $k$, when the state is $s$. A stationary policy $\pi$ is one for which $\mu_k = \mu_l \overset{\triangle}{=} \mu$ (a time-invariant function), $\forall k, l = 0, 1, \ldots$. Associated with any transition to a state $s'$ from a state $s$ under action $a$, is a 'single-stage' cost $g(s, a, s')$ where $g : S \times A \times S \to \mathcal{R}$ is called the cost function. The goal of the decision maker is to select actions $a_k, k \geq 0$ in response to the system states $s_k, k \geq 0$, observed one at a time, so as to minimize a long-term cost objective. We assume here that the number of states and actions is finite.

### 2.1 The Episodic or Stochastic Shortest Path Setting

We consider here the episodic or the stochastic shortest path problem where decision making terminates once a goal or terminal state is reached. We let $1, \ldots, p$ denote the set of non-terminal or regular states and $t$ be the terminal state. Thus, $S = \{1, 2, \ldots, p, t\}$ denotes the state space for this problem (Bertsekas, 2019).

Our basic setting here is similar to Chapter 3 of Bertsekas (2012) (see also Bertsekas (2019)), where it is assumed that under any policy there is a positive probability of hitting the goal state $t$ in at most $p$ steps starting from any initial (non-terminal) state, that would in turn signify that the problem would terminate in a finite though random amount of time.

Under a given deterministic policy $\pi$, define

$$V_\pi(s) = E_\pi \left[ \sum_{k=0}^{T} g(X_k, \mu_k(X_k), X_{k+1}) \mid X_0 = s \right], \tag{1}$$

where $T > 0$ is a finite random time at which the process enters the terminal state $t$. Here $E_\pi[\cdot]$ indicates that all actions are chosen according to policy $\pi$ depending on the system state at any instant. We assume that there is no action that is feasible in the state $t$ and the process terminates once it reaches $t$.

Let $\Pi$ denote the set of all admissible policies. The goal here is to find the optimal value function $V^*(i), i \in S$, where

$$V^*(i) = \min_{\pi \in \Pi} V_\pi(i) = V_{\pi^*}(i), \ i \in S, \tag{2}$$

with $\pi^*$ being the optimal policy. A related goal then would be to search for the optimal policy $\pi^*$. It turns out that in these problems, there exist stationary policies that are optimal, and so it is sufficient to restrict the search to the class of stationary policies.

A stationary policy $\pi$ is called a proper policy (cf. pp.174 of Bertsekas (2012)) if

$$\hat{p}_\pi \stackrel{\triangle}{=} \max_{s=1,\ldots,p} P(X_p \neq t \mid X_0 = s, \pi) < 1.$$

In other words, regardless of the initial state $s$, there is a positive probability of termination after at most $p$ stages when using a proper policy $\pi$ and moreover $P(T < \infty) = 1$ under such a policy. An admissible policy (and so also a stationary policy) can be randomized as well. A randomized admissible policy or simply a randomized policy is the sequence $\psi = \{\phi_0, \phi_1, \ldots\}$ with each $\phi_i : S \to P(A)$ being a distribution $\phi_i(s) = (\phi_i(s, a), a \in A(s))$ for the action to be chosen in the $i$th stage in state $s$. A stationary randomized policy is one for which $\phi_j = \phi_k \stackrel{\triangle}{=} \phi, \ \forall j, k = 0, 1, \ldots$. Here and in the rest of the paper, we shall assume that the policies are stationary randomized and are parameterized via a certain parameter $\theta \in C \subset \mathcal{R}^d$, a compact and convex set. We make the following assumption:

**Assumption 1** *All stationary randomized policies $\phi_\theta$ parameterized by $\theta \in C$ are proper.*

In practice, one might be able to relax this assumption (as with the model-based analysis of Bertsekas (2012)) by (a) assuming that for policies that are not proper, $V_\pi(i) = \infty$ for at least one non-terminal state $i$ and (b) there exists a proper policy. The optimal value function satisfies the Bellman equation: For $s = 1, \ldots, p$,

$$V^*(s) = \min_{a \in A(s)} \left( \bar{g}(s, a) + \sum_{j=1}^{p} p(j \mid s, a) V^*(j) \right), \tag{3}$$

where $\bar{g}(s, a) = \sum_{j=1}^{p} p(j|s, a) g(s, a, j) + p(t|s, a) g(s, a, t)$ is the expected single-stage cost in a non-terminal state $s$ when a feasible action $a$ is chosen. It can be shown, see Bertsekas (2012), that an optimal stationary proper policy exists.

## 2.2 The Policy Gradient Theorem

Policy gradient methods perform a gradient search within the prescribed class of parameterized policies. Let $\phi_\theta(s, a)$ denote the probability of selecting action $a \in A(s)$ when the state is $s \in S$ and the policy parameter is $\theta \in C$. We assume that $\phi_\theta(s, a)$ is continuously differentiable in $\theta$. A common example here is of the parameterized Boltzmann or softmax policies. Let $\phi_\theta(s) \stackrel{\triangle}{=} (\phi_\theta(s, a), a \in A(s)), \ s \in S$ and $\phi_\theta \stackrel{\triangle}{=} (\phi_\theta(s), s \in S)$.

We assume that trajectories of states and actions are available either as real data or from a simulation. Let $G_k = \sum_{j=k}^{T-1} g_j$ denote the sum of costs until termination (likely when a goal state is reached) on a trajectory starting from instant $k$. Note that if all actions are chosen according to a policy $\phi$, then the value and Q-value functions (under $\phi$) would be $V_\phi(s) = E_\phi[G_k \mid X_k = s]$ and $Q_\phi(s, a) = E_\phi[G_k \mid X_k = s, Z_k = a]$, respectively. In what follows, for ease of notation, we let $V_\theta \equiv V_{\phi_\theta}$ and $Q_\theta \equiv Q_{\phi_\theta}$, respectively.

The policy gradient theorem for episodic problems has the following form, cf. Chapter 13, pp.325, of Sutton & Barto (2018):

$$\nabla V_\theta(s_0) = \sum_{s \in S} \mu(s) \sum_{a \in A(s)} \nabla_\theta \pi(s, a) Q_\theta(s, a), \tag{4}$$

where $\mu(s), s \in S$, is defined as $\mu(s) = \dfrac{\eta(s)}{\sum_{s' \in S} \eta(s')}$ where $\eta(s) = \sum_{k=0}^{\infty} p^k(s|s_0, \phi_\theta)$, $s \in S$, with $p^k(s|s_0, \phi_\theta)$ being the $k$-step transition probability of going to state $s$ from $s_0$ under the policy $\phi_\theta$.

The REINFORCE algorithm (Sutton & Barto (2018); Williams (1992)) makes use of the expression in (4). In what follows, we present an alternative algorithm based on REINFORCE that incorporates one and two measurement (zeroth order) SF gradient estimators. Since our algorithm caters to episodic tasks, it performs updates whenever a certain prescribed recurrent state is visited, see Cao (2007); Marbach & Tsitsiklis (2001). We refer to our one-simulation (resp. two-simulation) algorithm as the One-SF-REINFORCE (SFR-1) (resp. Two-SF-REINFORCE (SFR-2)) algorithm.

## 3 The One-Simulation SF REINFORCE (SFR-1) Algorithm

We assume that data on the $n$th trajectory is represented in the form of the tuples $(s_k^n, a_k^n, g_k^n, s_{k+1}^n)$, $k = 0, 1, \ldots, T_n - 1$ with $T_n$ being the termination instant on the $n$th trajectory, $n \geq 1$. Also, $s_j^n$ is the state at instant $j$ in the $n$th trajectory. Further, $a_k^n$ and $g_k^n$ are the action chosen and the cost incurred, respectively, at instant $k$ in the $n$th trajectory. Let $\Gamma : \mathcal{R}^d \to C$ denote a projection operator that projects any $x = (x_1, \ldots, x_d)^T \in \mathcal{R}^d$ to its nearest point in $C$. For ease of exposition, we assume that $C$ is a $d$-dimensional rectangle having the form $C = \prod_{i=1}^{d} [c_{i,\min}, c_{i,\max}]$, where $-\infty < c_{i,\min} < c_{i,\max} < \infty$, $\forall i = 1, \ldots, d$. Then $\Gamma(x) = (\Gamma_1(x_1), \ldots, \Gamma_d(x_d))^T$ with $\Gamma_i : \mathcal{R} \to [c_{i,\min}, c_{i,\max}]$ such that $\Gamma_i(x_i) = \min(c_{i,\max}, \max(c_{i,\min}, x_i))$, $i = 1, \ldots, d$. Also, let $\mathcal{C}(C)$ denote the space of all continuous functions from $C$ to $\mathcal{R}^d$.

In what follows, we present a procedure that incrementally updates the parameter $\theta$. Let $\theta(n)$ denote the parameter value obtained after the $n$th update of this procedure which depends on the $n$th episode and which is run using the policy parameter $\Gamma(\theta(n-1) + \delta_{n-1}\Delta(n-1))$. Here, for $n \geq 0$, where $\theta(n) = (\theta_1(n), \ldots, \theta_d(n))^T \in \mathcal{R}^d$, $\delta_n > 0$ $\forall n$ with $\delta_n \to 0$ as $n \to \infty$ and $\Delta(n) = (\Delta_1(n), \ldots, \Delta_d(n))^T, n \geq 0$, where $\Delta_i(n), i = 1, \ldots, d, n \geq 0$ are independent random variables distributed according to the $N(0,1)$ distribution.

Algorithm (5) below is used to update the parameter $\theta \in C \subset \mathcal{R}^d$. Let $\chi^n$ denote the $n$th state-action trajectory $\chi^n = \{s_0^n, a_0^n, s_1^n, a_1^n, \ldots, s_{T-1}^n, a_{T-1}^n, s_T^n\}$, $n \geq 0$ where the actions $a_0^n, \ldots, a_{T-1}^n$ in $\chi^n$ are obtained using the policy parameter $\theta(n) + \delta_n \Delta(n)$. The instant $T$ denotes the termination instant in the trajectory $\chi^n$ and corresponds to the instant when the terminal or goal state $t$ is reached. Note that the various actions in the trajectory $\chi^n$ are chosen according to the policy $\phi_{(\theta(n)+\delta_n\Delta(n))}$. The initial state is assumed to be sampled from a given initial distribution $\nu = (\nu(i), i \in S)$ over states. Let $G^n = \sum_{k=0}^{T-1} g_k^n$ denote the sum of costs until termination on the trajectory $\chi^n$ with $g_k^n \equiv g(s_k^n, a_k^n, s_{k+1}^n)$. The update rule that we consider here is the following: For $n \geq 0, i = 1, \ldots, d$,

$$\theta_i(n+1) = \Gamma_i \left( \theta_i(n) - \alpha(n) \left( \Delta_i(n) \frac{G^n}{\delta_n} \right) \right). \tag{5}$$

**Assumption 2** *The step-size sequence $\{\alpha(n)\}$ satisfies $\alpha(n) > 0$, $\forall n$, $\sum_n \alpha(n) = \infty$, $\sum_n \left( \dfrac{\alpha(n)}{\delta_n} \right)^2 < \infty$.*

After the $(n-1)$st episode, $\theta(n)$ is computed using (5). The perturbed parameter $\theta(n) + \delta_n \Delta(n)$ is then obtained after sampling $\Delta(n)$ from the multivariate Gaussian distribution as explained previously and thereafter a new trajectory governed by this perturbed parameter is generated with the initial state in each episode sampled according to a given distribution $\nu$.

## 4 Variants for Improved Performance

We present here variants of this algorithm that result in improved bias and/or variance performance. We show the convergence results for all the algorithms and also test their performance empirically.

### 4.1 Two-Simulation SF REINFORCE (SFR-2) Algorithm

The idea here is to use two system simulations instead of one in order to reduce the estimator bias. As with SFR-1, we assume that we have access to trajectories of data that are used for performing the parameter updates. Let $\chi^{n+}$ and $\chi^{n-}$ denote two state-action trajectories or episodes generated after the $n$th update of the parameter. These correspond to $\chi^{n+} = \{s_0^{n+}, a_0^{n+}, s_1^{n+}, a_1^{n+}, \ldots, s_{T-1}^{n+}, a_{T-1}^{n+}, s_T^{n+}\}$, $n \geq 0$ where the actions $a_0^{n+}, \ldots, a_{T-1}^{n+}$ are obtained using the policy parameter $\theta(n) + \delta_n \Delta(n)$. Likewise, the actions $a_0^{n-}, \ldots, a_{T-1}^{n-}$ in $\chi^{n-}$ are obtained using the policy parameter $\theta(n) - \delta_n \Delta(n)$. As before, a new random vector $\Delta(n)$ is generated after $\theta(n)$ is obtained using the algorithm but the same $\Delta(n)$ is used in both the policy parameters used to generate the two trajectories. The initial state in both these episodes is independently sampled from the same initial distribution $\nu = (\nu(i), i \in S)$ over states. Let $G^{n+} = \sum_{k=0}^{T-1} g_k^{n+}$ denote the return or the sum of costs until termination on the trajectory $\chi^{n+}$, with $g_k^{n+} \equiv g(s_k^{n+}, a_k^{n+}, s_{k+1}^{n+})$. Similarly, we let $G^{n-} = \sum_{k=0}^{T-1} g_k^{n-}$ denote the return or the sum of costs until termination on the trajectory $\chi^{n-}$, with $g_k^{n-} \equiv g(s_k^{n-}, a_k^{n-}, s_{k+1}^{n-})$. The update rule that we consider here is the following: For $n \geq 0, i = 1, \ldots, d$,

$$\theta_i(n+1) = \Gamma_i \left( \theta_i(n) - \alpha(n) \left( \Delta_i(n) \frac{(G^{n+} - G^{n-})}{2\delta_n} \right) \right). \tag{6}$$

One may view (5)-(6) as zeroth order policy gradient algorithms involving one and two measurement gradient estimators where the performance gradient is estimated using direct function measurements.

**Lemma 1** *The gradient estimator in SFR-2 has a lower estimator bias than the one in SFR-1.*

*Proof:* We show the proof of this result in Appendix A.3. ∎

### 4.2 SF REINFORCE with Signed Updates

As expected and (also) reported in the literature (Sutton & Barto (2018)), REINFORCE typically suffers from high iterate-variance. We observe this problem even when SF-REINFORCE is used. To counter the problem of high iterate-variance, we use the sign function $sgn(\cdot)$ in the updates defined as follows: $sgn(x) = +1$ if $x > 0$ and $sgn(x) = -1$ otherwise.

#### 4.2.1 SFR-1 with Signed Updates

The update rule is exactly the same as (5) except that only the sign of the increment is used in the update: $\forall i = 1, \ldots, d$,

$$\theta_i(n+1) = \Gamma_i \left( \theta_i(n) - \alpha(n) sgn \left( \Delta_i(n) \frac{G^n}{\delta_n} \right) \right). \tag{7}$$

#### 4.2.2 SFR-2 with Signed Updates

As with the SFR-1 case, the update rule here is the same as (6) except that the update rule involves the sign of the update increment. Thus, we have, $\forall i = 1, \ldots, d$,

$$\theta_i(n+1) = \Gamma_i \left( \theta_i(n) - \alpha(n) sgn \left( \Delta_i(n) \frac{(G^{n+} - G^{n-})}{2\delta_n} \right) \right). \tag{8}$$

### 4.3 Two-Simulation SF REINFORCE with Clipped Updates

We present here updates obtained after using norm-wise (Zhang et al., 2020) or component-wise (Pascanu et al., 2013) clipping on the gradients. Norm clip $f_m : \mathbb{R}^d \to \mathbb{R}^d$ of a vector $x$ in $\mathbb{R}^d$ is defined as the projection of $x$ onto the ball of radius $m$, centered at the origin, i.e., $f_m(x) = \min(m, \|x\|)\frac{x}{\|x\|}$. Similarly, component-clip $f_c : \mathbb{R}^d \to \mathbb{R}^d$ of a vector $x$ in $\mathbb{R}^d$ is defined as the projection of $x$ onto the box centered at origin of side $2c$. That is, for $x = (x_1, \dots, x_d)$, we let $f_c(x) = (\max(\min(x_1, c), -c), \dots, \max(\min(x_d, c), -c))$.

Then, for $f \in \{f_c, f_m\}$, the updates will be of the form

$$\theta_i(n+1) = \Gamma_i\left(\theta_i(n) - \alpha(n)f\left(\Delta_i(n)\frac{(G^{n+} - G^{n-})}{2\delta_n}\right)\right). \tag{9}$$

We have the following basic result on variance of the signed as well as clipped updates.

**Lemma 2** *(i) Let $Y = sgn(X)$ be a random variable that is the sign of another random variable $X$. Then $\mathrm{Var}(Y) \leq 1$ regardless of $\mathrm{Var}(X)$.*

*(ii) Let $U$ be a random vector in $\mathbb{R}^d$, and $V = f(U) \in \mathbb{R}^d$, $f \in \{f_c, f_m\}$, then*

$$\mathrm{Tr}(\mathrm{Cov}V) := \mathrm{E}\|V - \mathrm{E}U\|^2 \leq \mathrm{E}\|U - \mathrm{E}U\|^2 := \mathrm{Tr}(\mathrm{Cov}U).$$

*Proof:* The proof of this result is given in Appendix A.4.

**Remark 1** *(i) It follows from Lemma 2(i) that $\mathrm{Var}\left(sgn\left(\Delta_i(n)\frac{G^n}{\delta_n}\right)\right) \leq 1$, $\forall\theta$, for Signed SFR-1 and similarly, $\mathrm{Var}\left(sgn\left(\Delta_i(n)\frac{(G^{n+} - G^{n-})}{2\delta_n}\right)\right) \leq 1$, $\forall\theta$, for Signed SFR-2. Notice that the estimators without the sign function, namely SFR-1 and SFR-2, are expected to have higher variance as $G^n$, $G^{n+}$ and $G^{n-}$ are the returns or sum of rewards on the trajectories that are then divided by a small quantity $\delta_n$. Clearly, unlike Signed SFR-1 and Signed SFR-2, one cannot provide a uniform bound on the variance of the estimators in SFR-1 and SFR-2 and their variance is expected to be much higher than the signed versions. This is also validated through our experiments.*

*(ii) It follows from Lemma 2(ii) that the total variance of the gradient, must decrease after projection in SF Reinforce with Clipped Updates. This is because*

$$\mathrm{TrCov}f\left(\Delta_i(n)\frac{(G^{n+} - G^{n-})}{2\delta_n}\right) \leq \mathrm{TrCov}\left(\Delta_i(n)\frac{(G^{n+} - G^{n-})}{2\delta_n}\right).$$

### 4.4 Evolutionary Strategies (ES) Algorithms

We recall the evolutionary strategies (ES) algorithms, see (Flaxman et al., 2005; Salimans et al., 2017; Mania et al., 2018a). There are two versions of this update rule that are popular in the literature. These are based on one and two simulation SF. We refer to these as ES-v1 and ES-v2, respectively, depending on whether the gradient estimator used is SFR-1 or SFR-2. Let $\Delta^m(n), m = 1, \dots, k$ be independent random vectors $\Delta^m(n) = (\Delta_1^m(n), \dots, \Delta_d^m(n))^T$ with the $\Delta_i^m(n), i = 1, \dots, d$, $m = 1, \dots, k$, $n \geq 0$ being i.i.d random variables with each having the distribution $N(0, 1)$.

#### 4.4.1 ES-v1

Let $\chi^{n,m}$, $m = 1, \dots, k$ denote $k$ state-action trajectories run with parameters $\theta(n) + \delta_n\Delta^m(n)$, $m = 1, \dots, k$, respectively, and $G^{n,m}$ denote the return on the $m$th trajectory starting from time 0. Here, $k \geq 1$ is a given fixed integer. The update rule then is as follows: For $i = 1, \dots, d$,

$$\theta_i(n+1) = \Gamma_i\left(\theta_i(n) - \alpha(n)\frac{1}{k\delta_n}\sum_{m=1}^{k}\Delta_i^m(n)G^{n,m}\right), \tag{10}$$

This algorithm requires $k$ function measurements for one update. The value of $k$ is chosen by the user. Note $k = 1$ corresponds to SFR-1 in this case.

### 4.4.2 ES-v2

Let $\chi^{n,m+}$ and $\chi^{n,m-}$, $m = 1, \ldots, k$ denote $2k$ state-action trajectories run with parameters $\theta(n) + \delta_n \Delta^m(n)$ and $\theta(n) - \delta_n \Delta^m(n)$, $m = 1, \ldots, k$, respectively. Let $G^{n,m+}$ and $G^{n,m-}$ denote the returns obtained on $\chi^{n,m+}$ and $\chi^{n,m-}$, respectively, starting from time 0. The update rule here is the following: For $i = 1, \ldots, d$,

$$\theta_i(n+1) = \Gamma_i \left( \theta_i(n) - \alpha(n) \frac{1}{2k\delta_n} \sum_{m=1}^{k} \Delta_i^m(n) \left( G^{n,m+} - G^{n,m-} \right) \right). \tag{11}$$

As before, $k$ is a priori chosen. The algorithm requires $2k$ function measurements for any given parameter update, and for $k = 1$, we recover the SFR-2 update. We present an asymptotic convergence analysis of ES-v1 and ES-v2.

**Remark 2** *The ARS algorithms of Mania et al. (2018a;b) that we implement, make use of best b out of k directions over which the above sample averages are taken, see Appendix B for details of ARS. We show the asymptotic analysis of the ES variants. The same for the ARS variants is not shown as it follows along the same lines as the ES algorithms with the average taken over b (best directions) instead of all k directions.*

## 5 Convergence Analysis

We present here the main convergence results for all the algorithms considered. The detailed proofs of all of these results are provided in Appendix A. The convergence analysis follows the ordinary differential equation (ODE) method. The iterates in all the algorithms are seen to be stable, i.e., uniformly bounded almost surely, because of the projection. The sequence of steps below are described for SFR-1 but similar steps are followed for the other algorithms as well.

We first show that the noisy increment term in each update rule can be written as the sum of a performance function (that is an expectation over the noisy sample function) and a zero-mean noise term that we show forms a martingale difference sequence over various iterations, see for instance, (12) and Lemma 3. We then show that the martingale sequence obtained by summing over the products of the martingale differences and the corresponding learning rates over recursions is almost surely convergent, for instance, see Lemma 6. We also show an important result namely that the parameter-dependent value function $V_\theta$ is differentiable abd its gradient is Lipschitz continuous satisfying a linear growth bound, see for instance, Lemma 4. The remainder of the proof then shows that the algorithm corresponds to a noisy Euler discretization of the associated ODE and so tracks the set of internally chain recurrent points of the ODE (see for instance, Theorem 1). We prove the asymptotic convergence of SFR-1 and SFR-2 under just two assumptions, namely Assumptions 1 and 2. In fact, we prove all the basic requirements such as the parameterized value function being differentiable with a Lipschitz continuous gradient (Lemma 4). This is unlike papers on ES/ARS (Malik et al., 2020) that make much stronger assumptions but do not prove whether these assumptions are valid in the settings that they consider.

Similar steps are followed for SFR-2 except that we also show that the bias in the gradient estimator for SFR-2 is smaller than the corresponding bias in SFR-1. This results in overall better accuracy of the procedure. In Section 5.3, we prove the convergence of Signed SFR-2. Our first observation towards proving convergence of such a scheme is that it can be written as a difference between two indicator functions whose expectation would be the probability of the mentioned event. We again show that the scheme tracks the stable fixed points of another associated ODE but which (the stable fixed points) are the same as those of the earlier ODE. We also show here that the variance of the iterates in this scheme is lower than in the previous recursion. Indeed the signed version of SFR type algorithms has not been previously studied.

Finally, for proving the convergence of the ES algorithms, we first define a suitable mean squared difference function whose gradient evaluated to zero results in the form of the corresponding ES gradient estimator that is a sample average over $k$ or $2k$ different function measurements depending on whether the algorithm used is ES-v1 or ES-v2. Suitable Taylor's expansions and the relevant properties on the perturbations then show that the said estimators indeed approximate the gradient in the expectation. Suitable martingale difference

sequences are then derived. The rest of the proof then follows using a similar set of arguments as for the other algorithms.

### 5.1 Convergence of SFR-1

The detailed proofs of the various results here are given in Appendix A.1.

We begin by rewriting the recursion (5) as follows:

$$\theta_i(n+1) = \Gamma_i\left(\theta_i(n) - \alpha(n)E\left[\Delta_i(n)\frac{G^n}{\delta_n}|\mathcal{F}_n\right] + M_{n+1}^i\right),\tag{12}$$

where $M_{n+1}^i = \Delta_i(n)\frac{G^n}{\delta_n} - E\left[\Delta_i(n)\frac{G^n}{\delta_n}|\mathcal{F}_n\right], n \geq 0$, with $\mathcal{F}_n \triangleq \sigma(\theta(m), m \leq n, \Delta(m), \chi^m, m < n), n \geq 1$, being a sequence of increasing sigma fields with $\mathcal{F}_0 = \sigma(\theta(0))$. Let $M_n \triangleq (M_n^1, \ldots, M_n^d)^T, n \geq 0$.

**Lemma 3** $(M_n, \mathcal{F}_n), n \geq 0$ *is a martingale difference sequence.*

**Proposition 1** *We have*

$$E\left[\Delta_i(n)\frac{G^n}{\delta_n} \mid \mathcal{F}_n\right] = \sum_{s \in S}\nu(s)\nabla_i V_{\theta(n)}(s) + o(\delta_n) \; a.s.$$

In the light of Proposition 1, we can rewrite (5) as follows:

$$\theta(n+1) = \Gamma(\theta(n) - \alpha(n)(\sum_s \nu(s)\nabla V_{\theta(n)}(s) + M_{n+1} + \beta(n))),\tag{13}$$

where $\beta(n) = (\beta_1(n), \ldots, \beta_d(n))^T$ with $\beta_i(n) = E\left[\Delta_i(n)\frac{G_n}{\delta} \mid \mathcal{F}_n\right] - \sum_s \nu(s)\nabla_i V_{\theta(n)}(s)$. From Proposition 1, it then follows that $\beta(n) = o(\delta_n)$.

**Lemma 4** *The function* $\nabla V_\theta(s)$ *is Lipschitz continuous in* $\theta$. *Further,* $\exists$ *a constant* $K_1 > 0$ *such that* $\| \nabla V_\theta(s) \| \leq K_1(1+ \| \theta \|)$.

**Lemma 5** *The sequence* $(M_n, \mathcal{F}_n), n \geq 0$ *satisfies* $E[\|M_{n+1}\|^2 \mid \mathcal{F}_n] \leq \frac{\hat{L}}{\delta_n^2}$, *for some constant* $\hat{L} > 0$.

Define now a sequence $Z_n, n \geq 0$ according to $Z_n = \sum_{m=0}^{n-1} a(m)M_{m+1}, n \geq 1$, with $Z_0 = 0$.

**Lemma 6** $(Z_n, \mathcal{F}_n), n \geq 0$ *is an almost surely convergent martingale sequence.*

Consider now the following ODE:

$$\dot{\theta}(t) = \bar{\Gamma}(-\sum_s \nu(s)\nabla V_\theta(s)),\tag{14}$$

where $\bar{\Gamma} : \mathcal{C}(C) \to \mathcal{C}(\mathcal{R}^d)$ is defined according to

$$\bar{\Gamma}(v(x)) = \lim_{\eta \to 0}\left(\frac{\Gamma(x + \eta v(x)) - x}{\eta}\right).\tag{15}$$

Let $H \triangleq \{\theta \mid \bar{\Gamma}(-\sum_s \nu(s)\nabla V_\theta(s)) = 0\}$ denote the set of all equilibria of (14). By Lemma 11.1 of Borkar (2022), the only possible $\omega$-limit sets that can occur as invariant sets for the ODE (14) are subsets of $H$. Let $\bar{H} \subset H$ be the set of all internally chain recurrent points of the ODE (14). Our main result below is based on Theorem 5.3.1 of Kushner & Clark (1978) for projected stochastic approximation algorithms. We state this theorem in Appendix A along with the assumptions needed there that we verify for our analysis.

**Theorem 1** *The iterates* $\theta(n), n \geq 0$ *governed by (5) converge almost surely to* $\bar{H}$.

## 5.2 Convergence of SFR-2

The proofs of the results below are given in Appendix A.2. The analysis proceeds in a similar manner here as for the one-simulation SF. Let

$$H_i(\theta(n), \Delta(n)) = \Delta_i(n) \left( \frac{V_{\theta(n)+\delta(n)\Delta(n)} - V_{\theta(n)-\delta(n)\Delta(n)}}{2\delta(n)} \right).$$

**Proposition 2**

$$E\left[ \Delta_i(n) \left( \frac{G^{n+} - G^{n-}}{2\delta_n} \right) \mid \mathcal{F}_n \right] = \sum_s \nu(s) E[H_i(\theta(n), \Delta(n))|\mathcal{F}_n] = \sum_{s \in S} \nu(s) \nabla_i V_{\theta(n)}(s) + o(\delta_n) \ a.s.$$

The main result on convergence of the stochastic recursions is the following:

**Theorem 2** *The iterates $\theta(n), n \geq 0$ governed by (7) converge almost surely to $\bar{H}$.*

## 5.3 Convergence of Signed SFR-2

We present here the convergence analysis of the two-simulation signed SF REINFORCE algorithm (or Signed SFR-2). The analysis of the one-simulation counterpart is analogous and hence is not provided.

Let $e_i(n) \overset{\triangle}{=} H_i(\theta(n), \Delta(n)) - \nabla_i V_{\theta(n)}$. Further, let $F_i(e|\theta) = P(e_i(n) \leq e|\theta(n) = \theta)$ be the conditional distribution of $e_i(n)$ given $\theta(n) = \theta$. We make the following assumptions:

(A1) $P(e_i(n) \leq e|\theta(m), m \leq n) = F_i(e|\theta(n))$ independent of $n$.

(A2) The maps $(e, \theta) \mapsto F_i(e|\theta)$ and $\theta \mapsto \nabla_i V_\theta$ are Lipschitz continuous.

(A3) For all $\theta$ and $i = 1, \dots, d$, $F_i(0|\theta) = 1/2$.

(A4) $\alpha(n) > 0, \forall n, \sum_n \alpha(n) = \infty, \sum_n \left( \frac{\alpha(n)}{\delta_n} \right)^2 < \infty.$

Consider the following ODE associated with the above recursion:

$$\dot{\theta}_i(t) = \bar{\Gamma}_i(-(1 - 2F_i(-\sum_s \nu(s)\nabla_i V_\theta(s)|\theta))), \ t \geq 0, \ i = 1, \dots, d. \tag{16}$$

For $x = (x_1, \dots, x_d)^T$, let $\bar{\Gamma}(x) = (\bar{\Gamma}_1(x_1), \dots, \bar{\Gamma}_d(x_d))^T$. Also, let $F(-\nabla V_\theta)|\theta) \overset{\triangle}{=} (F_1(-\sum_s \nu(s)\nabla_1 V_\theta(s)|\theta), \dots, F_d(-\sum_s \nu(s)\nabla_d V_\theta(s)|\theta))$ and let $K = \{\theta|(\bar{\Gamma}(-(1 - 2F(-\sum_s \nu(s)\nabla V_\theta(s)|\theta)) = 0\}$ denote the set of equilibria of (16). Further, let $\bar{K} \subset K \subset \{\theta|\bar{\Gamma}(\langle(1 - 2F(-\sum_s \nu(s)\nabla V_\theta(s)|\theta)), \sum_s \nu(s)\nabla V_\theta(s)\rangle) = 0\}$ denote the largest invariant set contained in $K$.

**Theorem 3 (Convergence of Signed SFR-2)** $\{\theta(n)\}$ *governed as per (8) converges as $n \to \infty$ almost surely to $\bar{K}$.*

*Proof:* The proof is given in Appendix A.5. ∎

**Remark 3** *Suppose $\theta \in K$ is such that $\theta$ is in the interior of the constraint set. Then, from Assumptions (A2)-(A3) and Theorem 3, $\sum_s \nu(s)\nabla V_\theta(s) = 0$. For $\theta$ on the boundary of the constraint set, either $\sum_s \nu(s)\nabla V_\theta(s) = 0$ or $\sum_s \nu(s)\nabla V_\theta(s) \neq 0$ but in the latter case, $\bar{\Gamma}(\sum_s \nu(s)\nabla V_\theta(s) = 0)$. The latter are spurious fixed points that occur at the boundary of the constraint set, see Kushner & Yin (1997).*

### 5.4 Convergence of Two-Simulation SF REINFORCE with Clipped Gradients

**Theorem 4 (Convergence of SFR-2 with Clipped Gradients)** $\{\theta(n)\}$ *governed as per (9) converges as* $n \to \infty$ *almost surely to* $\bar{K}$, *for each of the settings (i)* $f = f_c$ *and (ii)* $f = f_m$.

    *Proof:* The proof is given in Appendix A.6. ∎

### 5.5 Convergence of ES-v1

**Theorem 5 (Convergence of ES-v1)** *The iterates* $\theta(n)$, $n \geq 0$ *governed according to (10) converge almost surely to* $\bar{H}$ *as* $n \to \infty$.

    *Proof:* The proof is given in Appendix A.7. ∎

### 5.6 Convergence of ES-v2

**Theorem 6 (Convergence of ES-v2)** *The iterates* $\theta(n)$, $n \geq 0$ *governed according to (11) converge almost surely to* $\bar{H}$ *as* $n \to \infty$.

    *Proof:* The proof is available in Appendix A.8. ∎

## 6 Numerical Results

We show the results of experiments on various settings. The first set of experiments are on a simple 2D gridworld environment with different state sizes[1]. Subsequently, we show the results of experiments on four different continuous control MuJoCo environments[2].

### 6.1 Stochastic Gridworld Environment

### 6.1.1 Setup

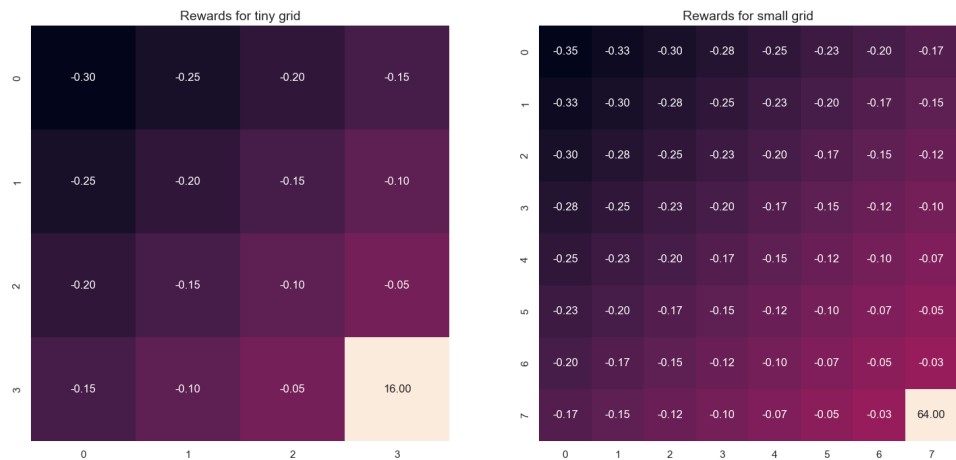

Figure 1: Stochastic Gridworld problem

The gridworld environment consists of an $L \times L$ grid where the agent starts in the top-left corner and aims to reach the terminal state at the bottom-right. Rewards are structured to penalize the agent based on the Manhattan distance to the goal, with penalties decreasing as the agent moves closer, as shown in Figure 1. The agent can move in all four cardinal directions, but action execution is stochastic: with probability

---

[1] https://github.com/deepakhr1999/smooth-functional-reinforce

[2] https://github.com/deepakhr1999/ARS-SFR, forked from modestyachts/ARS to implement SFR and compare with ARS.

$1 - p = 0.9$, the agent moves as intended, and with probability $p = 0.1$, it moves in a perpendicular direction, introducing uncertainty in the transition dynamics. An episode terminates upon reaching the goal or after a maximum number of steps to ensure bounded returns. Full details of the environment configuration, hyperparameters, and policy architectures for each algorithm are provided in Appendix C.

### 6.1.2 Results

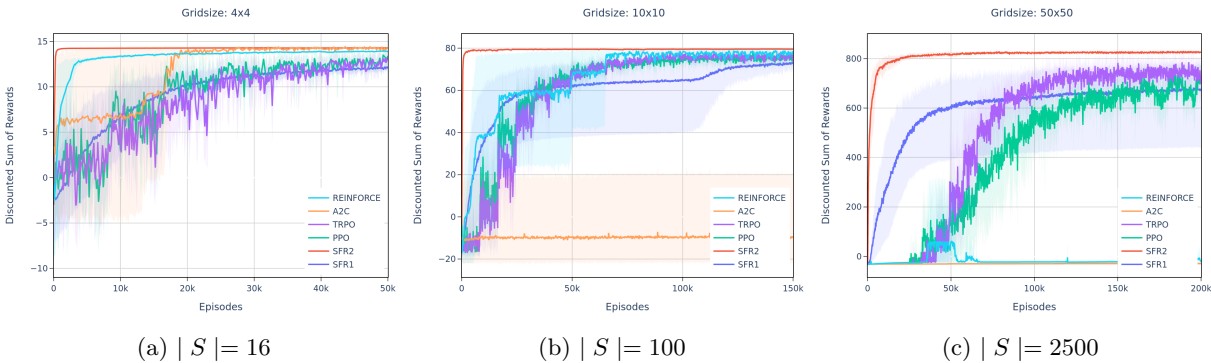

(a) $\mid S \mid = 16$        (b) $\mid S \mid = 100$        (c) $\mid S \mid = 2500$

Figure 2: Plots showing performance of iterates of algorithms on various grid sizes.

| Gridsize | SFR-1 | **SFR-2** | PPO | TRPO | A2C | REINFORCE |
|---|---|---|---|---|---|---|
| 4x4 | $12.12 \pm 0.5$ | $\mathbf{14.28 \pm 0.0}$ | $12.72 \pm 0.2$ | $12.48 \pm 0.1$ | $14.20 \pm 0.0$ | $13.90 \pm 0.1$ |
| 10x10 | $72.66 \pm 3.6$ | $\mathbf{79.53 \pm 0.1}$ | $75.38 \pm 0.3$ | $75.77 \pm 0.3$ | $-9.60 \pm 29.7$ | $76.90 \pm 4.9$ |
| 50x50 | $677.55 \pm 233.1$ | $\mathbf{824.96 \pm 5.6}$ | $682.97 \pm 12.9$ | $741.81 \pm 10.7$ | $-28.75 \pm 5.4$ | $-18.79 \pm 8.4$ |

Table 1: Total reward (higher is better)(mean $\pm$ standard error) of algorithms on different grid sizes.

| Gridsize / Algo | Thresholds | SFR-1 | **SFR-2** | PPO | TRPO | A2C | REINFORCE |
|---|---|---|---|---|---|---|---|
| 4x4 | 10 | 19071 | **164** | 16554 | 20889 | 16654 | 1863 |
| 10x10 | 60 | 30380 | **344** | 30098 | 33656 | NaN | 28837 |
| 50x50 | 600 | 42680 | **2177** | 98462 | 74272 | NaN | NaN |

Table 2: Steps taken (less is better) by average training curves of algorithms to cross reward thresholds on grids of various sizes. "NaN" means the algorithm never achieved this cumulative reward.

We evaluate the performance of six algorithms: SFR-1, SFR-2, REINFORCE, A2C, PPO, and TRPO on Gridworld environments of varying sizes. For each grid size, we report the best results obtained across different hyperparameter settings, averaging outcomes over 10 random seeds and running each algorithm for a fixed number of interactions (see Appendix C.1 for details). As expected, SFR-2 consistently achieves the best performance and exhibits the lowest variance across all grid sizes, as seen in Table 1. From Table 2, it also learns the optimal policy with the fewest number of steps. PPO and TRPO follow closely, demonstrating competitive performance with low variance. SFR-1, while effective, suffers from higher variance due to the bias introduced by one-sided updates, validating Lemma 1. REINFORCE and A2C perform reasonably well on smaller grids (4×4 and 10×10), but struggle to scale to larger environments such as the 50×50 grid. The training curves can be seen in Figure 2. Across 10 seeds, we plot the average reward as the thick line and shade the standard deviation around it. Overall, SFR-2 emerges as the most reliable and sample-efficient method in this setting, highlighting its potential for fast and stable learning in online reinforcement learning scenarios. For our subsequent experiments, we therefore just show performance comparisons of the various algorithms with SFR-2.

## 6.2 MuJoCo Locomotion

### 6.2.1 Setup

We evaluate first the performance of SFR-2 and its signed and clipped variants on MuJoCo locomotion tasks (Todorov et al., 2012), comparing them to the ARS (Augmented Random Search) algorithm (Mania et al., 2018b). We also subsequently show comparisons with other well known algorithms, namely TRPO, PPO and A2C. Beyond convergence, we focus on how efficiently each algorithm uses environment interactions.

We choose to include ARS for two reasons. Firstly, they show that with just linear policies, one could achieve competitive results even in such complex tasks. Secondly, ARS is very similar to our SFR-2 algorithm, but with a few key differences. A detailed comparison of ARS with our SFR-2 algorithm, along with the full ARS pseudo-code is provided in Appendix B. Essentially, ARS employs multiple workers to evaluate $2 \times k$ perturbed policies along $k$ directions and selects the top $b$ directions to estimate a gradient, which is then normalized using the standard deviation of returns. Thus, ARS consumes $2kH$ environment interactions per update, where $H$ is the horizon size (considered constant for the purpose of this argument). In contrast, SFR-2 simplifies this by setting $k = b = 1$ and normalizes the gradient using the standard deviation of perturbations, resulting in the consumption of only $2H$ interactions per update.

While ARS leverages a greater number of trajectories per update to reduce variance in its gradient estimates, this leads to fewer updates under a fixed interaction budget. In contrast, SFR-2 updates its policy more frequently, albeit with noisier gradients due to limited trajectory usage per update. For a fair comparison, we fix the total number of environment interactions (Table 3) and evaluate performance under that constraint. To address this trade-off in our SFR-2, we investigate the extent to which variance-control techniques such as gradient clipping and signed updates can mitigate instability and enhance performance across both algorithms.

### 6.2.2 Comparison with Augmented Random Search (ARS)

We conduct experiments with various hyperparameter settings: $k$, $b$, $\alpha$, and $\nu$. Initially, optimization algorithms are run using a grid search over all hyperparameter combinations with a single seed per combination. From this, the best-performing hyperparameter configuration is selected and further evaluated across four additional seeds (five seeds in total). Table 4 reports the mean $\pm$ standard deviation for these five seeds. The hyperparameter grids, along with the optimal hyperparameters for each task can be found in Appendix C.2.

| Task | State Dimension | Action Dimension | Max Timesteps |
|------|----------------:|-----------------:|--------------:|
| Swimmer | 8 | 2 | 10,000,000 |
| Hopper | 11 | 3 | 30,000,000 |
| HalfCheetah | 17 | 6 | 50,000,000 |
| Walker2d | 17 | 6 | 50,000,000 |

Table 3: Task specifications used in our experiments. For each environment, we report the dimensionality of the state and action spaces, along with the maximum number of timesteps used for training. State vectors lie in $\mathbb{R}^d$, while each component of the action vector is bounded within the interval $[-1, 1]$.

From Table 4, as expected, the unmodified ARS-v1t and ARS-v2t algorithms outperform the unmodified SFR-2 across most tasks. However, when modifications such as Component_Clip, Norm_Clip, and Signed Update are introduced, the variance in policy updates is reduced. This is particularly beneficial for SFR-2, which performs a larger number of updates within a fixed environment interaction budget. As a result, SFR-2 shows notable performance gains. On the Swimmer and HalfCheetah tasks, it consistently outperforms both ARS variants across all three modifications. For Hopper, the modified versions of SFR-2 surpass ARS-v1t but remain slightly behind ARS-v2t. In the Walker2d environment, SFR-2 with Component_Clip outperforms ARS-v2t, though it still trails ARS-v1t. Overall, these results suggest that SFR-2 benefits more from the introduced variance-control techniques, making it highly competitive with ARS when the number of interactions with the environment is held fixed across algorithms.

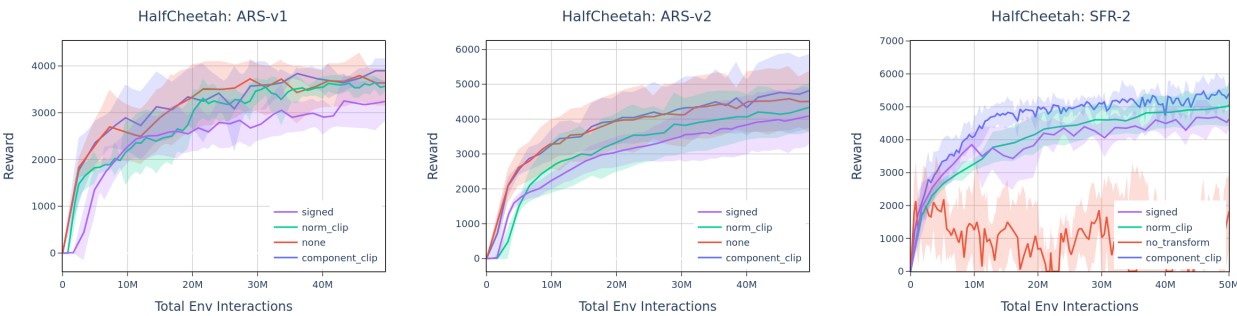

Figure 3: Comparison of different variants on the HalfCheetah environment.

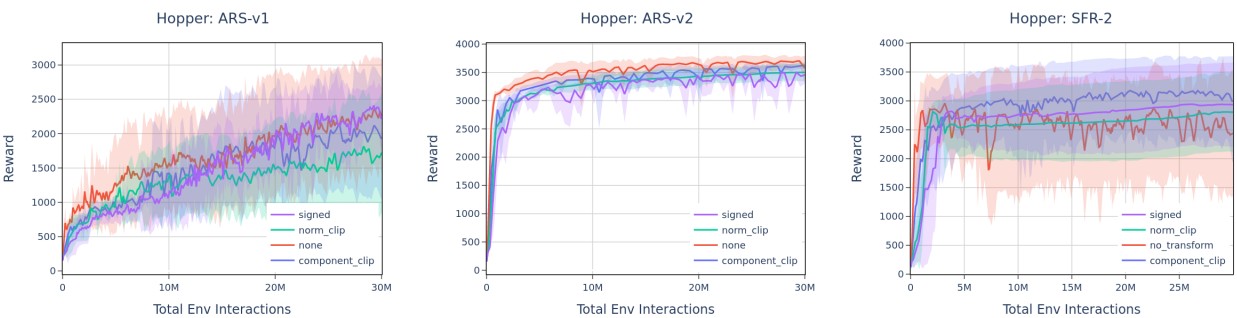

Figure 4: Comparison of different variants on the Hopper environment.

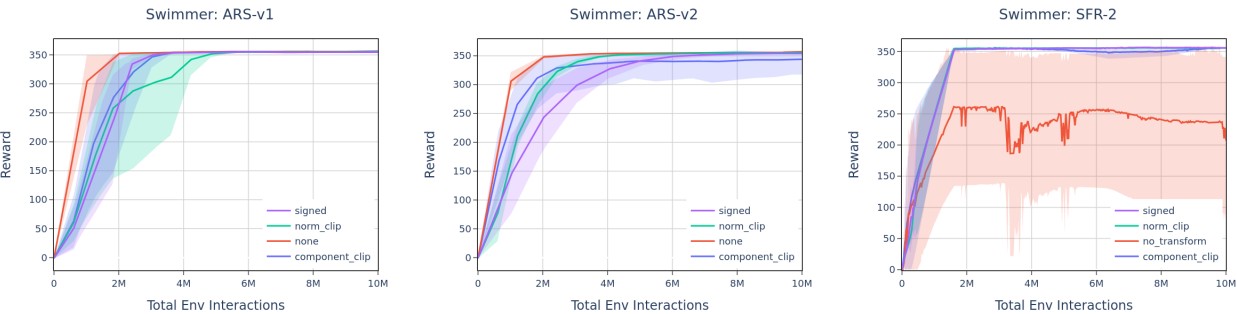

Figure 5: Comparison of different variants on the Swimmer environment.

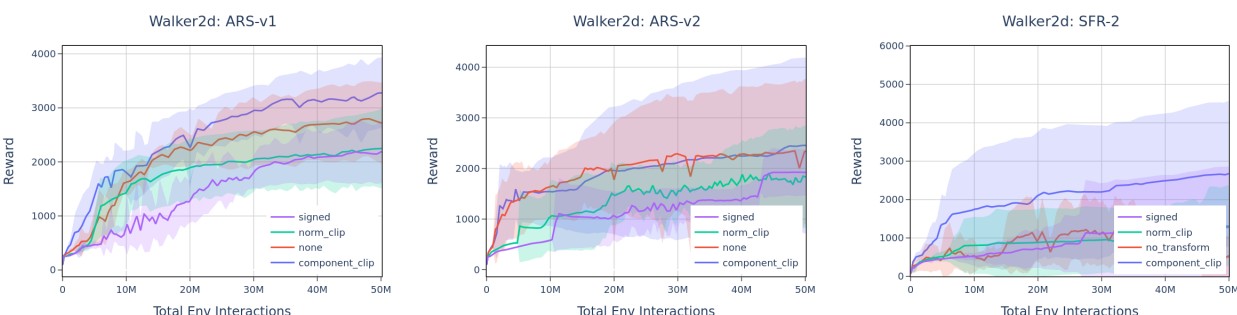

Figure 6: Comparison of different variants on the Walker2d environment.

| Task | Algo used | Algo with Component_Clip | Original Algo | Algo with Norm_Clip | Algo with Signed Update |
|------|-----------|--------------------------|---------------|---------------------|-------------------------|
| Swimmer | ARS-v1t | $356.83 \pm 0.35$ | $356.82 \pm 0.80$ | $356.52 \pm 0.54$ | $356.10 \pm 0.56$ |
| | ARS-v2t | $345.57 \pm 27.36$ | $357.60 \pm 1.17$ | $356.54 \pm 0.55$ | $354.95 \pm 2.39$ |
| | **SFR-2 (ours)** | $357.19 \pm 1.85$ | $268.55 \pm 121.01$ | $357.4 \pm 1.26$ | $\mathbf{357.89 \pm 1.36}$ |
| HalfCheetah | ARS-v1t | $4097.17 \pm 156.33$ | $3889.75 \pm 601.84$ | $3786.04 \pm 140.15$ | $3345.06 \pm 518.20$ |
| | ARS-v2t | $4849.18 \pm 1094.69$ | $4621.09 \pm 908.41$ | $4377.46 \pm 724.93$ | $4110.96 \pm 846.69$ |
| | **SFR-2 (ours)** | $\mathbf{5762.27 \pm 499.75}$ | $2977.42 \pm 929.95$ | $5082.51 \pm 605.8$ | $5042.62 \pm 341.89$ |
| Hopper | ARS-v1t | $2312.74 \pm 817.09$ | $2603.87 \pm 591.21$ | $1924.58 \pm 752.77$ | $2612.73 \pm 307.14$ |
| | **ARS-v2t** | $3639.82 \pm 52.60$ | $\mathbf{3719.36 \pm 110.70}$ | $3511.56 \pm 151.94$ | $3518.42 \pm 103.51$ |
| | SFR-2 (ours) | $3256.43 \pm 698.7$ | $3215.54 \pm 757.42$ | $3123.01 \pm 422.79$ | $3194.95 \pm 410.85$ |
| Walker2d | **ARS-v1t** | $\mathbf{3354.91 \pm 567.67}$ | $2822.74 \pm 700.24$ | $2295.12 \pm 687.07$ | $2234.53 \pm 578.97$ |
| | ARS-v2t | $2652.87 \pm 1594.69$ | $2689.41 \pm 1121.71$ | $2179.55 \pm 810.47$ | $2028.48 \pm 441.79$ |
| | SFR-2 (ours) | $2718.52 \pm 1875.69$ | $1640.73 \pm 1072.84$ | $1382.85 \pm 1051.86$ | $1310.45 \pm 1560.97$ |

Table 4: Average reward and standard error performance on each task (across 5 seeds), constraining total environment interactions as per Table 3.

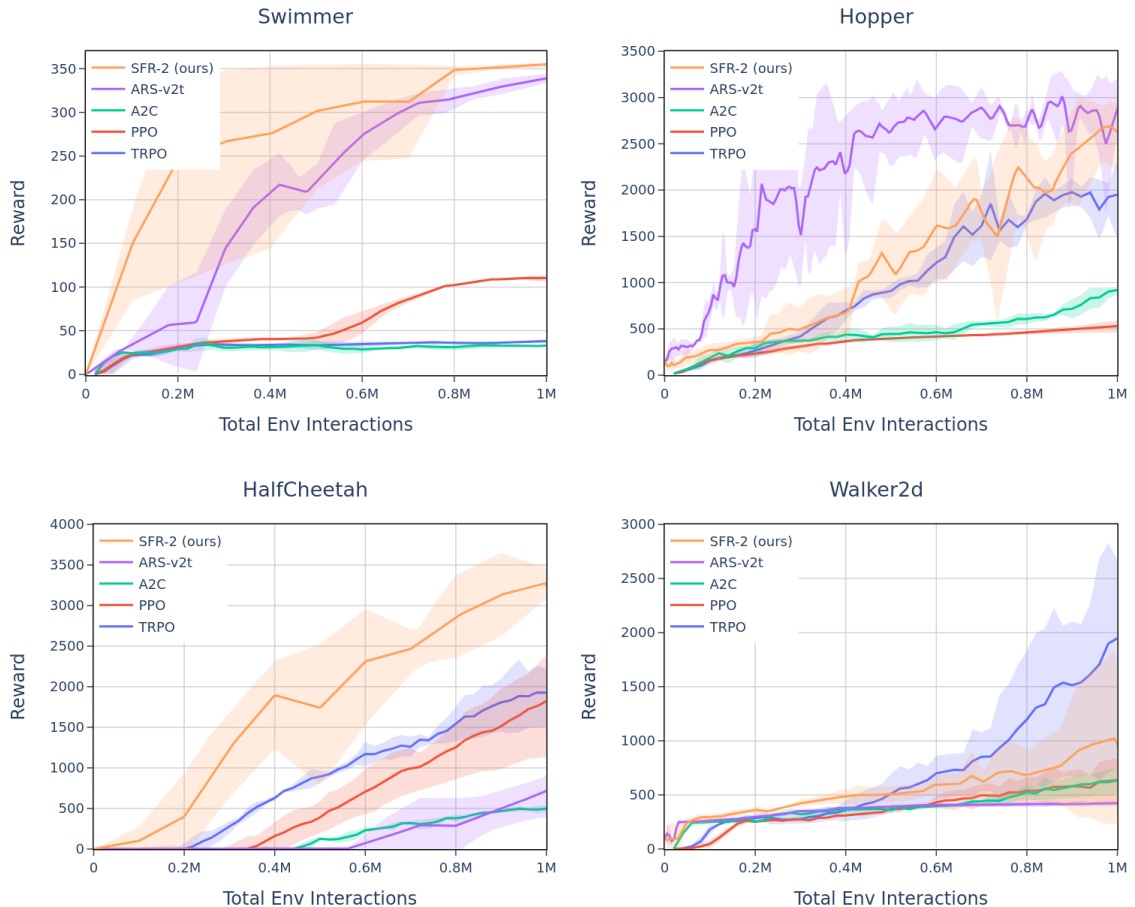

Figure 7: Performance plots across different environments.

### 6.2.3 Comparison with other benchmark algorithms

We compare now SFR-2 against ARS-v2t, TRPO, A2C, and PPO on MuJoCo tasks over a fixed budget of 1 million environment steps. For SFR-2 and ARS-v2t, we use the same hyperparameter grid from Appendix C.2. Thus, the best performing variants of both SFR-2 and ARS-v2t are used for comparison. For

the other baseline algorithms, we use the implementation and default hyperparameters provided in Ji et al. (2024). Our results are broadly consistent with prior findings in Mania et al. (2018a). Table 5 summarizes the final performance, and Figure 7 shows the learning curves.

SFR-2 achieves the highest reward in two of the four environments (Swimmer and HalfCheetah), and performs competitively and consistently across the remaining tasks.

| task | SFR-2 (ours) | ARS-v2t | PPO | A2C | TRPO |
|---|---|---|---|---|---|
| Swimmer | $355.37 \pm 2.21$ | $340.61 \pm 4.71$ | $111.29 \pm 2.45$ | $39.32 \pm 4.79$ | $39.02 \pm 1.05$ |
| Hopper | $2749.79 \pm 428.79$ | $3164.38 \pm 112.17$ | $530.77 \pm 69.75$ | $924.99 \pm 54.49$ | $2234.73 \pm 156.64$ |
| HalfCheetah | $3392.27 \pm 289.96$ | $810.55 \pm 396.24$ | $1824.04 \pm 686.48$ | $523.73 \pm 26.0$ | $1962.55 \pm 453.94$ |
| Walker2d | $1040.4 \pm 792.8$ | $428.32 \pm 20.25$ | $636.83 \pm 201.27$ | $643.65 \pm 119.77$ | $2023.76 \pm 1258.39$ |

Table 5: Average reward and standard error performance (across three seeds), constraining total environment interactions to be 1 million steps.

## 7  Conclusions

We presented model-free smoothed functional algorithms as suitable Monte-Carlo based alternatives to REINFORCE for the setting of episodic tasks. We also presented the clipped and signed variants of the algorithms and analysed the convergence of all the presented algorithms. We showed detailed empirical results of our algorithms, first on a simple gridworld problem before scaling them on the MuJoCo locomotion tasks. We showed performance comparisons with other algorithms, in particular, the ARS algorithms that have been investigated recently. For a fixed number of environment interactions, our algorithms are competitive against ARS and in fact their signed and clipped variants are superior to ARS on more than half the settings tried. With the incorporation of variance reduction techniques, SFR-2 establishes itself as a strong candidate algorithm for online learning settings that demand low per-update cost and rapid learning. As future work, it would be of interest to theoretically study the asymptotic rate of convergence results of the algorithms presented here. Such results for all algorithms including ES/ARS are not currently available.

## Acknowledgements

The authors wish to thank the Editor and anonymous reviewers for their comments that helped improve the overall quality of this paper. This work was supported in part by Project No. DFTM/02/3125/M/04/AIR-04 from DRDO under DIA-RCOE, by a J. C. Bose Research Grant with No. ANRF/JBG/2025/000209/HAA from ANRF, Government of India, and by the Walmart Centre for Tech Excellence, Indian Institute of Science, Bengaluru.

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

# A  Details of the Convergence Analysis

We present here the details of the convergence analysis and give the proofs of the various results. We begin first with the results for the One-Simulation SF REINFORCE algorithm. We will subsequently sketch the analysis of the two-simulation SF algorithm. Finally, we shall discuss the convergence analysis of the algorithms with signed updates.

## A.1  Convergence of SFR-1

**Proof of Lemma 3:** Notice that

$$M_n^i = \Delta_i(n-1)\frac{G^{n-1}}{\delta_{n-1}} - E\left[\Delta_i(n-1)\frac{G^{n-1}}{\delta_{n-1}} \mid \mathcal{F}_{n-1}\right].$$

The first term on the RHS above is clearly measurable $\mathcal{F}_n$ while the second term is measurable $\mathcal{F}_{n-1}$ and hence measurable $\mathcal{F}_n$ as well. Further, from Assumption 1, each $M_n$ is integrable. Finally, it is easy to verify that

$$E[M_{n+1}^i \mid \mathcal{F}_n] = 0, \ \forall i.$$

The claim follows.

**Proof of Proposition 1:** Note that

$$E\left[\Delta_i(n)\frac{G^n}{\delta_n} \mid \mathcal{F}_n\right] = E\left[E\left[\Delta_i(n)\frac{G^n}{\delta_n} \mid \mathcal{G}_n\right] \mid \mathcal{F}_n\right],$$

where $\mathcal{G}_n \triangleq \sigma(\theta(m), \Delta(m), m \leq n, \chi^m, m < n), n \geq 1$ is a sequence of increasing sigma fields with $\mathcal{G}_0 = \sigma(\theta(0), \Delta(0))$. It is clear that $\mathcal{F}_n \subset \mathcal{G}_n, \forall n \geq 0$. Now,

$$E\left[\Delta_i(n)\frac{G^n}{\delta_n} \mid \mathcal{G}_n\right] = \frac{\Delta_i(n)}{\delta_n}E[G^n \mid \mathcal{G}_n].$$

Let $s_0^n = s$ denote the initial state in the trajectory $\chi^n$. Recall that the initial state $s$ is chosen randomly from the distribution $\nu$. Thus,

$$E[G^n \mid \mathcal{G}_n] = \sum_s \nu(s)E[G^n \mid s_0^n = s, \phi_{\theta(n)+\delta_n\Delta(n)}]$$

$$= \sum_s \nu(s)V_{\theta(n)+\delta_n\Delta(n)}(s).$$

Thus, with probability one,

$$E\left[\Delta_i(n)\frac{G^n}{\delta_n} \mid \mathcal{G}_n\right] = \sum_s \nu(s)\left(\Delta_i(n)\frac{V_{\theta(n)+\delta_n\Delta(n)}(s)}{\delta_n}\right).$$

Hence, it follows almost surely that

$$E\left[\Delta_i(n)\frac{G^n}{\delta_n} \mid \mathcal{F}_n\right] = \sum_s \nu(s)E\left[\Delta_i(n)\frac{V_{\theta(n)+\delta_n\Delta(n)}(s)}{\delta_n} \mid \mathcal{F}_n\right].$$

Using a Taylor's expansion of $V_{\theta(n)+\delta_n\Delta(n)}(s)$ around $\theta(n)$ gives us

$$V_{\theta(n)+\delta_n\Delta(n)}(s_n) = V_{\theta(n)}(s_n) + \delta_n\Delta(n)^T\nabla V_{\theta(n)}(s_n) + \frac{\delta_n^2}{2}\Delta(n)^T\nabla^2 V_{\theta(n)}(s_n)\Delta(n) + o(\delta_n^2).$$

Now recall that $\Delta(n) = (\Delta_i(n), i = 1, \ldots, d)^T$. Thus,

$$\Delta(n)\frac{V_{\theta(n)+\delta_n\Delta(n)}(s_n)}{\delta_n} = \frac{1}{\delta_n}\Delta(n)V_{\theta(n)}(s_n)$$

$$+\Delta(n)\Delta(n)^T \nabla V_{\theta(n)}(s_n)$$

$$+\frac{\delta_n}{2}\Delta(n)\Delta(n)^T \nabla^2 V_{\theta(n)}(s_n)\Delta(n) + o(\delta_n).$$

Now observe from the properties of $\Delta_i(n), \forall i, n$, that
(i) $E[\Delta(n)] = 0$ (the zero-vector), $\forall n$, since $\Delta_i(n) \sim N(0,1)$, $\forall i, n$.
(ii) $E[\Delta(n)\Delta(n)^T] = I$ (the identity matrix), $\forall n$.

(iii) $E\left[\sum_{i,j,k=1}^{d} \Delta_i(n)\Delta_j(n)\Delta_k(n)\right] = 0.$

Property (iii) follows from the facts that (a) $E[\Delta_i(n)\Delta_j(n)\Delta_k(n)] = 0$, $\forall i \neq j \neq k$, (b) $E[\Delta_i(n)\Delta_j^2(n)] = 0$, $\forall i \neq j$ (this pertains to the case where $i \neq j$ but $j = k$ above) and (c) $E[\Delta_i^3(n)] = 0$ (for the case when $i = j = k$ above). These properties follow from the independence of the random variables $\Delta_i(n)$, $i = 1, \dots, d$ and $n \geq 0$, as well as the fact that they are all distributed $N(0,1)$. The claim now follows from (i)-(iii) above.

Recall that from Proposition 1, it follows that $\beta(n) = o(\delta_n)$.

**Proof of Lemma 4**: It can be seen from (4) that $V_\theta(s)$ is continuously differentiable in $\theta$. It can also be shown as in Theorem 3 of Furmston et al. (2016) that $\nabla^2 V_\theta(s)$ exists and is continuous. Since $\theta$ takes values in $C$, a compact set, it follows that $\nabla^2 V_\theta(s)$ is bounded and thus $\nabla V_\theta(s)$ is Lipschitz continuous.

Finally, let $L_1^s > 0$ denote the Lipschitz constant for the function $\nabla V_\theta(s)$. Then, for a given $\theta_0 \in C$,

$$\| \nabla V_\theta(s) \| - \| \nabla V_{\theta_0}(s) \| \leq \| \nabla V_\theta(s) - \nabla V_{\theta_0}(s) \|$$

$$\leq L_1^s \| \theta - \theta_0 \|$$

$$\leq L_1^s \| \theta \| + L_1^s \| \theta_0 \| .$$

Thus, $\| \nabla V_\theta(s) \| \leq \| \nabla V_{\theta_0}(s) \| + L_1^s \| \theta_0 \| + L_1^s \| \theta \|$ . Let $K_s \triangleq \| \nabla V_{\theta_0}(s)\| + L_1^s\|\theta_0\|$ and $K_1 \triangleq \max(K_s, L_1^s, s \in S)$. Thus, $\| \nabla V_\theta(s) \| \leq K_1(1+ \| \theta \|)$. Note here that since $|S| < \infty$, $K_1 < \infty$ as well. The claim follows.

**Proof of Lemma 5:** Note that

$$\|M_{n+1}\|^2 = \sum_{i=1}^{d}(M_{n+1}^i)^2$$

$$= \sum_{i=1}^{d}\left(\Delta_i^2(n)\frac{(G^n)^2}{\delta_n^2} + \frac{1}{\delta_n^2}E\left[\Delta_i(n)G^n \mid \mathcal{F}_n\right]^2\right.$$

$$\left. -2\Delta_i(n)\frac{G^n}{\delta_n^2}E\left[\Delta_i(n)G^n \mid \mathcal{F}_n\right]\right).$$

Thus,

$$E[\|M_{n+1}\|^2 \mid \mathcal{F}_n] = \frac{1}{\delta_n^2}\sum_{i=1}^{d}\left(E[\Delta_i^2(n)(G^n)^2 \mid \mathcal{F}_n]\right.$$

$$\left. -E^2[\Delta_i(n)G^n \mid \mathcal{F}_n]\right).$$

The claim now follows from Assumption 1 and the fact that all single-stage costs are bounded (cf. pp.174, Chapter 3 of Bertsekas (2012)).

**Proof of Lemma 6:** It is easy to see that $Z_n$ is $\mathcal{F}_n$-measurable $\forall n$. Further, it is integrable for each $n$ and moreover $E[Z_{n+1} \mid \mathcal{F}_n] = Z_n$ almost surely since $(M_{n+1}, \mathcal{F}_n)$, $n \geq 0$ is a martingale difference sequence by

Lemma 3. It is also square integrable from Lemma 5. The quadratic variation process of this martingale will be convergent almost surely if

$$\sum_{n=0}^{\infty} E[\|Z_{n+1} - Z_n\|^2 \mid \mathcal{F}_n] < \infty \text{ a.s.} \tag{17}$$

Note that

$$E[\|Z_{n+1} - Z_n\|^2 \mid \mathcal{F}_n] = \alpha(n)^2 E[\|M_{n+1}\|^2 \mid \mathcal{F}_n].$$

Thus,

$$\sum_{n=0}^{\infty} E[\|Z_{n+1} - Z_n\|^2 \mid \mathcal{F}_n] = \sum_{n=0}^{\infty} \alpha(n)^2 E[\|M_{n+1}\|^2 \mid \mathcal{F}_n]$$

$$\leq \hat{L} \sum_{n=0}^{\infty} \left( \frac{\alpha(n)}{\delta_n} \right)^2,$$

by Lemma 5. (17) now follows as a consequence of Assumption 2. Now $(Z_n, \mathcal{F}_n)$, $n \geq 0$ can be seen to be convergent from the martingale convergence theorem for square integrable martingales Borkar (1995).

Our main result below is based on Theorem 5.3.1 of Kushner & Clark (1978) for projected stochastic approximation algorithms. Before we proceed further, we recall that result below.

Let $C \subset \mathcal{R}^d$ be a compact and convex set as before and $\Gamma : \mathcal{R}^d \to C$ denote the projection operator that projects any $x = (x_1, \ldots, x_d)^T \in \mathcal{R}^d$ to its nearest point in $C$.

Consider now the following the $d$-dimensional stochastic recursion

$$X_{n+1} = \Gamma(X_n + \alpha(n)(h(X_n) + \xi_n + \beta_n)), \tag{18}$$

under the assumptions listed below. Also, consider the following ODE associated with (18):

$$\dot{X}(t) = \bar{\Gamma}(h(X(t))). \tag{19}$$

As mentioned before, let $\mathcal{C}(C)$ denote the space of all continuous functions from $C$ to $\mathcal{R}^d$. The operator $\bar{\Gamma} : \mathcal{C}(C) \to \mathcal{C}(\mathcal{R}^d)$ is defined according to (15). The limit in (15) exists and is unique since $C$ is a convex set (the way it is defined). If $C$ is not convex and the limit is not unique, one may consider the set of all limit points of (15). Note also that from its definition, $\bar{\Gamma}(v(x)) = v(x)$ if $x \in C^o$ (the interior of $C$). This is because for such an $x$, one can find $\eta > 0$ sufficiently small so that $x + \eta v(x) \in C^o$ as well and hence $\Gamma(x + \eta v(x)) = x + \eta v(x)$. On the other hand, if $x \in \partial C$ (the boundary of $C$) is such that $x + \eta v(x) \notin C$, for any small $\eta > 0$, then $\bar{\Gamma}(v(x))$ is the projection of $v(x)$ to the tangent space of $\partial C$ at $x$.

Consider now the assumptions listed below.

(B1) The function $h : \mathcal{R}^d \to \mathcal{R}^d$ is continuous.

(B2) The step-sizes $\alpha(n), n \geq 0$ satisfy

$$\alpha(n) > 0 \forall n, \ \sum_n \alpha(n) = \infty, \ \alpha(n) \to 0 \text{ as } n \to \infty.$$

(B3) The sequence $\beta_n, n \geq 0$ is a bounded random sequence with $\beta_n \to 0$ almost surely as $n \to \infty$.

(B4) There exists $T > 0$ such that $\forall \epsilon > 0$,

$$\lim_{n \to \infty} P \left( \sup_{j \geq n} \max_{t \leq T} \left| \sum_{i=m(jT)}^{m(jT+t)-1} a(i)\xi_i \right| \geq \epsilon \right) = 0.$$

(B5) The ODE (19) has a compact subset $K$ of $\mathcal{R}^N$ as its set of asymptotically stable equilibrium points.

Let $t(n), n \geq 0$ be a sequence of positive real numbers defined according to $t(0) = 0$ and for $n \geq 1$, $t(n) = \sum_{j=0}^{n-1} a(j)$. By Assumption (B2), $t(n) \to \infty$ as $n \to \infty$. Let $m(t) = \max\{n \mid t(n) \leq t\}$. Thus, $m(t) \to \infty$ as $t \to \infty$. Assumptions (B1)-(B3) correspond to A5.1.3-A5.1.5 of Kushner & Clark (1978) while (B4)-(B5) correspond to A5.3.1-A5.3.2 there.

(Kushner & Clark, 1978, Theorem 5.3.1 (pp. 191-196)) essentially says the following:

**Theorem 7 (Kushner and Clark Theorem:)** *Under Assumptions (B1)–(B5), almost surely, $X_n \to K$ as $n \to \infty$.*

Finally, we come to the proof of our main result.

**Proof of Theorem 1:** In lieu of the foregoing, we rewrite (5) according to

$$\theta_i(n+1) = \Gamma_i\Big(\theta_i(n) - \alpha(n)\sum_s \nu(s)\nabla_i V_{\theta(n)}(s)$$

$$-\alpha(n)\beta_i(n) + M_{n+1}^i\Big), \tag{20}$$

where $\beta_i(n)$ is as in (13). We shall proceed by verifying Assumptions (B1)-(B5) and subsequently appeal to Theorem 5.3.1 of Kushner & Clark (1978) (i.e., Theorem 1 above) to claim convergence of the scheme. Note that Lemma 4 ensures Lipschitz continuity of $\nabla V_\theta(s)$ implying (B1). Next, from (B2), since $\delta_n \to 0$, it follows that $\alpha(n) \to 0$ as $n \to \infty$. Thus, Assumption (B2) holds as well. Now from Lemma 4, it follows that $\sum_s \nu(s)\nabla V_\theta(s)$ is uniformly bounded since $\theta \in C$, a compact set. Assumption (B3) is now verified from Proposition 1. Since $C$ is a convex and compact set, Assumption (B4) holds trivially. Finally, Assumption (B5) is also easy to see as a consequence of Lemma 6. Now note that for the ODE (14), $F(\theta) = \sum_s \nu(s)V_\theta(s)$ serves as an associated Lyapunov function and in fact

$$\nabla F(\theta)^T \bar{\Gamma}(-\sum_s \nu(s)\nabla V_\theta(s))$$

$$= (\sum_s \nu(s)\nabla_\theta V_\theta(s))^T \bar{\Gamma}(-\sum_s \nu(s)\nabla V_\theta(s)) \leq 0.$$

For $\theta \in C^o$ (the interior of $C$), it is easy to see that $\bar{\Gamma}(-\sum_s \nu(s)\nabla V_\theta(s)) = -\sum_s \nu(s)\nabla V_\theta(s)$, and

$$\nabla F(\theta)^T \bar{\Gamma}(-\sum_s \nu(s)\nabla V_\theta(s)) \quad < \quad 0 \text{ if } \theta \in H^c \cap C$$

$$= \quad 0 \text{ o.w.}$$

For $\theta \in \delta C$ (the boundary of $C$), there can additionally be spurious attractors, see Kushner & Yin (1997), that are also contained in $H$. The claim now follows from Theorem 5.3.1 of Kushner & Clark (1978).

## A.2 Convergence of SFR-2

The analysis proceeds in a similar manner as for the one-simulation SF except with $\dfrac{G^{n+} - G^{n-}}{2\delta_n}$ in place of $\dfrac{G^n}{\delta_n}$.

**Proof of Proposition 2:**

A similar calculation as with the proof of Proposition 1 would show that

$$E\left[\Delta_i(n)\left(\frac{G^{n+} - G^{n-}}{2\delta_n}\right) \mid \mathcal{F}_n\right] = \sum_s \nu(s)E\left[\Delta_i(n)\frac{(V_{\theta(n)+\delta_n\Delta(n)}(s) - V_{\theta(n)-\delta_n\Delta(n)}(s))}{2\delta_n} \mid \mathcal{F}_n\right].$$

Using Taylor's expansions of $V_{\theta(n)+\delta_n\Delta(n)}(s)$ and $V_{\theta(n)-\delta_n\Delta(n)}(s)$ around $\theta(n)$ gives us

$$\Delta(n)\left(\frac{V_{\theta(n)+\delta_n\Delta(n)}(s_n)-V_{\theta(n)-\delta_n\Delta(n)}(s_n)}{2\delta_n}\right) = \Delta(n)\Delta(n)^T\nabla V_{\theta(n)}(s_n) + o(\delta_n).$$

The zero order and second order terms directly cancel above instead of being zero-mean, thereby resulting in lower gradient estimator bias. The rest follows from properties (i)-(iii) mentioned previously for the one-simulation gradient SF. In particular, $E[\Delta(n)\Delta(n)^T] = I$.

**Proof of Theorem 2:**

In the light of Proposition 2, the proof here follows in a similar manner as one-simulation SF.

### A.3   Proof that SFR-2 has Lower Bias than SFR-1

Proof of Lemma 1

We show as part of the proof of Proposition 1 that

$$E\left[\Delta_i(n)\frac{G^n}{\delta_n} \mid \theta(n)\right] = \sum_s \nu(s)E\left[\Delta_i(n)\frac{V_{\theta(n)+\delta_n\Delta(n)}(s)}{\delta_n} \mid \theta(n)\right]. \tag{21}$$

Using a Taylor's expansion of $V_{\theta(n)+\delta_n\Delta(n)}(s)$ around $\theta(n)$ gives us

$$\frac{V_{\theta(n)+\delta_n\Delta(n)}(s_n)}{\delta_n} = \frac{V_{\theta(n)}(s_n)}{\delta_n} + \Delta(n)^T\nabla V_{\theta(n)}(s_n) + \frac{\delta_n}{2}\Delta(n)^T\nabla^2 V_{\theta(n)}(s_n)\Delta(n) + o(\delta_n).$$

Now recall that $\Delta(n) = (\Delta_i(n), i = 1, \ldots, d)^T$. Thus,

$$\Delta(n)\frac{V_{\theta(n)+\delta_n\Delta(n)}(s_n)}{\delta_n} = \frac{1}{\delta_n}\Delta(n)V_{\theta(n)}(s_n)+\Delta(n)\Delta(n)^T\nabla V_{\theta(n)}(s_n)+\frac{\delta_n}{2}\Delta(n)\Delta(n)^T\nabla^2 V_{\theta(n)}(s_n)\Delta(n)+o(\delta_n). \tag{22}$$

Taking now the conditional expectation as required in the RHS of (21), it can be seen that

$$E\left[\Delta_i(n)\frac{V_{\theta(n)+\delta_n\Delta(n)}(s)}{\delta_n} \mid \theta(n)\right] = \nabla_i V_{\theta(n)}(s_n) + O(\delta_n).$$

Now in the SFR-2 case, we require one more Taylor's expansion, namely of $V_{\theta(n)-\delta_n\Delta(n)}$ around the point $\theta(n)$. Here, like (22), one obtains

$$\Delta(n)\frac{V_{\theta(n)-\delta_n\Delta(n)}(s_n)}{\delta_n} = \frac{1}{\delta_n}\Delta(n)V_{\theta(n)}(s_n)-\Delta(n)\Delta(n)^T\nabla V_{\theta(n)}(s_n)+\frac{\delta_n}{2}\Delta(n)\Delta(n)^T\nabla^2 V_{\theta(n)}(s_n)\Delta(n)+o(\delta_n). \tag{23}$$

As part of the proof of Proposition 2, we observe as with Proposition 1 that

$$E\left[\Delta_i(n)\left(\frac{G^{n+}-G^{n-}}{2\delta_n}\right) \mid \mathcal{F}_n\right] = \sum_s \nu(s)E\left[\Delta_i(n)\frac{(V_{\theta(n)+\delta_n\Delta(n)}(s)-V_{\theta(n)-\delta_n\Delta(n)}(s)}{2\delta_n} \mid \mathcal{F}_n\right]. \tag{24}$$

From (22) and (23), one then gets

$$E\left[\Delta_i(n)\frac{(V_{\theta(n)+\delta_n\Delta(n)}(s)-V_{\theta(n)-\delta_n\Delta(n)}(s)}{2\delta_n} \mid \mathcal{F}_n\right] = \nabla_i V_{\theta(n)}(s_n) + o(\delta_n).$$

The important difference to note between the Taylor's expansions in the case of SFR-1 and SFR-2 is that in SFR-2, there is a direct cancellation of the bias terms $\frac{1}{\delta_n}\Delta(n)V_{\theta(n)}(s_n)$ and $\frac{\delta_n}{2}\Delta(n)\Delta(n)^T\nabla^2 V_{\theta(n)}(s_n)\Delta(n)$ that does not happen in SFR-1. The second term above does not contribute as much to the bias as the first term because the latter term has $\delta_n$ in the denominator that is expected to be small, in fact, $\delta_n \to 0$ as $n \to \infty$. This term averages out to zero eventually in SFR-1. In SFR-2, this term simply does not exist. This results in lower bias in SFR-2 as opposed to SFR-1 and eventually results in improved performance of SFR-2 over SFR-1.

### A.4 Proof of Lower Variance in the Signed and Clipped Variants

**Proof of Lemma 2:**

(i) Note from definition, $Y^2 = 1$, thereby $\mathrm{E}[Y^2] = 1$ and $0 \leq \mathrm{E}[Y]^2 \leq 1$. Thus, $\mathrm{Var}(Y) \leq 1$.

(ii) Recall $f \in \{f_c, f_m\}$ is a projection map from $\mathbb{R}^d$ to $C \subset \mathbb{R}^d$, where $C$ is a compact and convex set. We first show that this map is nonexpansive. In other words, we show that

$$\|f(x) - f(y)\| \leq \|x - y\|, \ \forall x, y \in \mathbb{R}^d.$$

Note that since $C$ is convex and compact,

$$\langle x - f(x), z - f(x) \rangle \leq 0, \ \forall z \in C.$$

Now since $f(y) \in C$, we have

$$\langle x - f(x), f(y) - f(x) \rangle \leq 0. \tag{25}$$

Similarly, we also have

$$\langle y - f(y), f(x) - f(y) \rangle \leq 0.$$

Changing the sign in both terms above gives us

$$\langle f(y) - y, f(y) - f(x) \rangle \leq 0. \tag{26}$$

Adding (25) and (26) gives

$$\langle x - f(x) + f(y) - y, f(y) - f(x) \rangle \leq 0.$$

In other words,

$$\langle x - y, f(y) - f(x) \rangle + \langle f(y) - f(x), f(y) - f(x) \rangle \leq 0.$$

Thus,

$$\|f(y) - f(x)\|^2 \leq \langle y - x, f(y) - f(x) \rangle$$
$$\leq \|y - x\| \|f(y) - f(x)\|,$$

by the Cauchy-Schwarz inequality. It now follows that

$$\|f(y) - f(x)\| \leq \|y - x\|.$$

Thus, $\mathrm{E}\|U - \mathrm{E}U\|^2 \geq \mathrm{E}\|f(U) - f(\mathrm{E}U)\|^2$. The claim now follows since $\mathrm{E}f(U) = \mathrm{argmin}_t \mathrm{E}\|f(U) - t\|^2$.
$\square$

### A.5 Convergence of Signed SFR-2

Recall that we have

$$H_i(\theta(n), \Delta(n)) = \Delta_i(n) \left[ \frac{V_{\theta(n) + \delta(n)\Delta(n)} - V_{\theta(n) - \delta(n)\Delta(n)}}{2\delta(n)} \right].$$

As explained previously,

$$E[H_i(\theta(n), \Delta(n)) | \mathcal{F}_n] = \nabla_i V_{\theta(n)} + o(\delta(n)).$$

Also, recall the 'error' in the $i$th component is given by

$$e_i(n) = H_i(\theta(n), \Delta(n)) - \nabla_i V_{\theta(n)} = \sum_{j \neq i} \Delta_i(n) \Delta_j(n) \nabla_j V_{\theta(n)} + o(\delta(n)).$$

**Proof of Theorem 3:**

We rewrite the algorithm as follows:

$$\theta_i(n+1) = \Gamma_i(\theta_i(n) - \alpha(n)sgn(H_i(\theta(n), \Delta(n))))$$

$$= \Gamma_i(\theta_i(n) - \alpha(n)(I(H_i(\theta(n), \Delta(n)) > 0) - I(H_i(\theta(n), \Delta(n)) \le 0))),$$

where $I(\cdot)$ is the indicator function. Thus, we have

$$\theta_i(n+1) = \Gamma_i(\theta_i(n) - \alpha(n)(1 - 2I(H_i(\theta(n), \Delta(n)) \le 0)))$$

$$= \Gamma_i(\theta_i(n) - a(n)(1 - 2P(H_i(\theta(n), \Delta(n)) \le 0|\mathcal{F}_n) + M_i(n+1))),$$

where

$$M_i(n+1) = 2P(H_i(\theta(n), \Delta(n)) \le 0|\mathcal{F}_n) - 2I(H_i(\theta(n), \Delta(n)) \le 0),$$

$$= 2P(e_i(n) \le -\sum_s \nu(s)\nabla_i V_{\theta(n)}(s)|\mathcal{F}_n) - 2I(e_i(n) \le -\sum_s \nu(s)\nabla_i V_{\theta(n)}(s))$$

$$= 2P(e_i(n) \le -\sum_s \nu(s)\nabla_i V_{\theta(n)}(s)|\theta(n)) - 2I(e_i(n) \le -\sum_s \nu(s)\nabla_i V_{\theta(n)}(s)),$$

by (A1). It is easy to see that $(M_i(n), \mathcal{F}_n), n \ge 0$ is a martingale difference sequence. Since $\sup_n |M_i(n)| \le 1$, and under (A4), it follows from an application of the martingale convergence theorem that $\sum_{m=0}^{n-1} a(m)M_{m+1}, n \ge 1$ is an almost surely convergent martingale.

It is easy to verify that $W(\theta) = \sum_s \nu(s)V_\theta(s)$ itself is a Lyapunov function for the ODE (16) since

$$\frac{dW(\theta)}{dt} = -\bar{\Gamma}(\langle (1 - 2F(-\sum_s \nu(s)\nabla V_\theta(s)|\theta)), \sum_s \nu(s)\nabla V_\theta(s)\rangle)$$

$$= -\sum_{i=1}^{N} \bar{\Gamma}_i((1 - 2F_i(-\sum_s \nu(s)\nabla_i V_\theta(s)|\theta)) \sum_s \nu(s)\nabla_i V_\theta(s).$$

From (A3), if $\sum_s \nu(s)\nabla_i V_\theta(s) > 0$, $(1 - 2F_i(-\sum_s \nu(s)\nabla_i V_\theta(s)|\theta)) \ge 0$ and $\frac{dW(\theta)}{dt} \le 0$. Similarly, if $\nabla_i V_\theta < 0$, $(1 - 2F_i(-\nabla_i V_\theta|\theta)) \le 0$ and $\frac{dV_\theta}{dt} \le 0$. Further, when $\sum_s \nu(s)\nabla_i V_\theta(s) = 0$, $\frac{dW(\theta)}{dt} = 0$. From Lasalle's invariance theorem in conjunction with Theorem 2 of Chapter 2 of Borkar (2022), it follows that $\theta(n), n \ge 0$ converges almost surely to the largest invariant set $\bar{K} \subset K \subset \{\theta|\bar{\Gamma}(\langle(1 - 2F(-\sum_s \nu(s)\nabla V_\theta(s)|\theta)), \sum_s \nu(s)\nabla V_\theta(s)\rangle) = 0\}$. The claim follows.

## A.6 Convergence of SFR-2 with Clipped Gradients

Proof of Theorem 4

Recall that the projection operator $f \in \{f_c, f_m\}$. Further, both $f = f_c$ and $f = f_m$ are continuous functions. Thus, observe that $f(\hat{\nabla} V_\theta) \to f(\nabla V_\theta)$, $\forall \theta$, where $\hat{\nabla} V_{\theta(n)}$ denotes the gradient estimate obtained from the two-simulation SF.

Following the same sequence of steps as in Theorem 2, it can be seen that the underlying ODE tracked by the algorithm is

$$\dot{\theta}(t) = \bar{\Gamma}(-\sum_s \nu(s)f(\nabla V_\theta(s))). \tag{27}$$

Note also that by construction in either case, namely (i) $f = f_c$ and (ii) $f = f_m$, we have that $f(\nabla V_\theta(s)) = 0$ if and only if $\nabla V_\theta(s) = 0$.

## A.7 Convergence of ES-v1

Proof of Theorem 5

Denote by $F(y)$ the mean squared difference function in variable $y \in \mathbb{R}^d$ defined as below:

$$F(y) = \frac{1}{2} E\left[\left(\frac{V_{\theta(n)+\delta_n \Delta(n)}(s)}{\delta_n} - y^T \Delta(n)\right)^2 |\theta(n)\right].$$

Then,

$$\nabla_y F(y) = E\left[-\left(\frac{V_{\theta(n)+\delta_n \Delta(n)}(s)}{\delta_n} - y^T \Delta(n)\right)\Delta(n)|\theta(n)\right].$$

Note now that

$$E[(y^T \Delta(n))\Delta(n)|\theta(n)] = E[\Delta(n)\Delta(n)^T y|\theta(n)] = y,$$

since $E[\Delta(n)\Delta(n)^T] = I$ (the identity matrix). Equating $\nabla_y F(y)$ to zero gives us upon simplification

$$y = E\left[\frac{V_{\theta(n)+\delta_n \Delta(n)}(s)}{\delta_n} \Delta(n)|\theta(n)\right],$$

thereby resulting in the gradient estimate

$$\hat{y}(n) = \frac{1}{k\delta_n} \sum_{m=1}^{k} V_{\theta(n)+\delta_n \Delta^m(n)} \Delta^m(n),$$

where the $\Delta^m(n), m = 1, \ldots, k$ are independent, having the multivariate Gaussian distribution with mean 0 and covariance matrix $I$. One may write

$$\hat{y}(n) = (\hat{y}_1(n), \ldots, \hat{y}_d(n))^T,$$

where

$$\hat{y}_i(n) = \frac{1}{k\delta_n} \sum_{m=1}^{k} V_{\theta(n)+\delta_n \Delta^m(n)} \Delta_i^m(n),$$

$i = 1, \ldots, d$. Thus, in the ES procedure, instead of using one sample of multivariate Gaussian, one calls $k$ samples of the same and takes the sample average of these. It can be seen that using a Taylor's expansion of $V_{\theta(n)+\delta_n \Delta^m(n)}(s)$ around $\theta(n)$ gives us

$$V_{\theta(n)+\delta_n \Delta^m(n)}(s_n) = V_{\theta(n)}(s_n) + \delta_n \Delta^m(n)^T \nabla V_{\theta(n)}(s_n) + \frac{\delta_n^2}{2} \Delta(n)^T \nabla^2 V_{\theta(n)}(s_n)\Delta(n) + o(\delta_n^2).$$

Now recall that $\Delta^m(n) = (\Delta_i^m(n), i = 1, \ldots, d)^T$. Thus,

$$\Delta^m(n)\frac{V_{\theta(n)+\delta_n \Delta^m(n)}(s_n)}{\delta_n} = \frac{1}{\delta_n}\Delta^m(n)V_{\theta(n)}(s_n)$$

$$+\Delta^m(n)\Delta^m(n)^T \nabla V_{\theta(n)}(s_n)$$

$$+\frac{\delta_n}{2}\Delta^m(n)\Delta^m(n)^T \nabla^2 V_{\theta(n)}(s_n)\Delta(n) + o(\delta_n).$$

Now observe from the properties of $\Delta_i^m(n), \forall m, i, n$, that
(i) $E[\Delta^m(n)] = 0$ (the zero-vector), $\forall n, \forall m = 1, \ldots, k$, since $\Delta_i^m(n) \sim N(0,1), \forall i, n, m$.
(ii) $E[\Delta^m(n)\Delta^m(n)^T] = I$ (the identity matrix), $\forall n, \forall m = 1, \ldots, k$.

(iii) $E\left[\sum_{i,j,k=1}^{d} \Delta_i^m(n)\Delta_j^m(n)\Delta_k^m(n)\right] = 0.$

Property (iii) follows from the facts that (a) $E[\Delta_i^m(n)\Delta_j^m(n)\Delta_l^m(n)] = 0, \forall i \neq j \neq l, \forall m = 1, \ldots, k$, (b)

$E[\Delta_i^m(n)(\Delta_j^m)^2(n)] = 0$, $\forall i \neq j$, $\forall m = 1, \ldots, k$ (this pertains to the case where $i \neq j$ but $j = k$ above for any given $m = 1, \ldots, k$) and (c) $E[(\Delta_i^m)^3(n)] = 0$ (for the case when $i = j = l$ above). These properties follow from the independence of the random variables $\Delta_i^m(n)$, $i = 1, \ldots, d$, $m = 1, \ldots, k$ and $n \geq 0$, as well as the fact that they are all distributed $N(0,1)$. As with Proposition 1, it can be seen that

$$E[\hat{y}(n) \mid \theta(n)] = \nabla V_{\theta(n)}(s) + O(\delta_n).$$

Now rewrite (10) as

$$\theta_i(n+1) = \Gamma_i\left(\theta_i(n) - \alpha(n)E[\hat{y}_i(n)|\theta(n)] - \alpha(n)M_i(n) - \alpha(n)N_i(n)\right) \tag{28}$$

where

$$M_i(n) = \hat{y}_i(n) - E[\hat{y}_i(n)|\theta(n)]$$

and

$$N_i(n) = \frac{1}{k\delta_n} \sum_{m=1}^{k} \Delta_i^m(n)(G^{n,m} - V_{\theta(n)+\delta_n\Delta^m(n)}),$$

As in Proposition 1, it can also be seen that $N_i(n), n \geq 0$ is a martingale difference sequence under the sequence of sigma algebras $\mathcal{G}_n, n \geq 1$, redefined as under: $\mathcal{G}_n \triangleq \sigma(\theta(l), \Delta^m(l), l \leq n, \chi^{l,m} l < n, m = 1, \ldots, k), n \geq 1$ is a sequence of increasing sigma fields with $\mathcal{G}_0 = \sigma(\theta(0), \Delta(0))$. The rest of the proof now follows via an application of the Kushner-Clark lemma as with SFR-1.

## A.8 Convergence of ES-v2

As with the case of ES-v1, define in this case

$$F(y) = \frac{1}{2}E\left[\left(\frac{V_{\theta(n)+\delta_n\Delta(n)}(s) - V_{\theta(n)-\delta_n\Delta(n)}(s)}{2\delta_n} - y^T\Delta(n)\right)^2 |\theta(n)\right].$$

Finding now $\nabla F(y)$ and setting it to zero gives us

$$y = E\left[\left(\frac{V_{\theta(n)+\delta_n\Delta(n)}(s) - V_{\theta(n)-\delta_n\Delta(n)}(s)}{2\delta_n}\right)\Delta(n)|\theta(n)\right],$$

resulting in the gradient estimate

$$\hat{y}(n) = \frac{1}{2k\delta_n} \sum_{m=1}^{k} (V_{\theta(n)+\delta_n\Delta^m(n)} - V_{\theta(n)-\delta_n\Delta^m(n)})\Delta(m),$$

and as with Proposition 2 using a similar sequence of steps as in Theorem 5 above, we obtain

$$E[\hat{y}(n) \mid \theta(n)] = \nabla V_{\theta(n)}(s) + o(\delta_n).$$

Note that one obtains $o(\delta_n)$ above as against $O(\delta_n)$ in a similar expansion in Theorem 5. This is because of the use of two function measurements at each epoch as opposed to one and which results in a direct cancellation of the first term on the RHS of the Taylor's expansions of $V_{\theta(n)+\delta_n\Delta^m(n)}$ and $V_{\theta(n)-\delta_n\Delta^m(n)}$ around $\theta(n)$, see Proposition 2. The rest of the proof now follows along the same lines as Theorem 5.

## B Augmented Random Search (ARS) and Comparisons

We first describe the ARS algorithm from (Mania et al., 2018a;b) and the various versions of it in detail as we incorporate this for purposes of comparison.

The ARS paper (Mania et al., 2018a) has ARS-v1 and ARS-v2, as given in the above algorithm. It uses only two-sided measurements since they result in more stable gradient estimates. Note from here on, in terms of

---

**Algorithm 1** Augmented Random Search (): **four versions V1, V1-t, V2 and V2-t**

---

1: **Hyperparameters:** step-size $\alpha$, number of directions sampled per iteration $k$, standard deviation of the exploration noise $\nu$, number of top-performing directions to use $b$ ($b < k$ is allowed only for **V1-t** and **V2-t**)

2: **Initialize:** $M_0 = \mathbf{0} \in \mathbb{R}^{p \times n}$, $\mu_0 = \mathbf{0} \in \mathbb{R}^n$, and $\Sigma_0 = \mathbf{I}_n \in \mathbb{R}^{n \times n}$, $j = 0$.

3: **while** ending condition not satisfied **do**

4:     Sample $\delta_1, \delta_2, \ldots, \delta_k$ in $\mathbb{R}^{p \times n}$ with i.i.d. standard normal entries.

5:     Collect $2k$ rollouts of horizon $H$ and their corresponding rewards using the $2k$ policies

$$\textbf{V1:} \quad \begin{cases} \pi_{j,l,+}(x) = (M_j + \nu\delta_l)x \\ \pi_{j,l,-}(x) = (M_j - \nu\delta_l)x \end{cases}$$

$$\textbf{V2:} \quad \begin{cases} \pi_{j,l,+}(x) = (M_j + \nu\delta_l)\operatorname{diag}(\Sigma_j)^{-1/2}(x - \mu_j) \\ \pi_{j,l,-}(x) = (M_j - \nu\delta_l)\operatorname{diag}(\Sigma_j)^{-1/2}(x - \mu_j) \end{cases}$$

    for $l \in \{1, 2, \ldots, k\}$.

6:     Sort the directions $\delta_l$ by $\max\{r(\pi_{j,l,+}), r(\pi_{j,l,-})\}$, denote by $\delta_{(l)}$ the $l$-th largest direction, and by $\pi_{j,(l),+}$ and $\pi_{j,(l),-}$ the corresponding policies.

7:     Make the update step:

$$M_{j+1} = M_j + \frac{\alpha}{b\sigma_R} \sum_{l=1}^{b} \left[ r(\pi_{j,(l),+}) - r(\pi_{j,(l),-}) \right] \delta_{(l)},$$

    where $\sigma_R$ is the standard deviation of the $2b$ rewards used in the update step.

8:     **V2** : Set $\mu_{j+1}$, $\Sigma_{j+1}$ to be the mean and covariance of the $2kH(j+1)$ states encountered from the start of training.[3]

9:     $j \leftarrow j + 1$

10: **end while**

.

---

ARS and our experiments, we do not deal with the one-sided version at all. Only two-sided is used. When we say reward, we refer to the total return of the episode. This is as per the terminology used in the original paper.

The authors use $M_j$ for the weights of the linear policy at iteration $j$. They sample $N$ perturbations and perturb the policy in two ways ($+$ and $-$) and obtain the total episodic reward against each perturbed policy (total of $2 \times N$ episodes are run through). As we read through the algorithm, the policy function of our SFR-1 and SFR-2 are similar to ARS-v1 from the line 5 of the algorithm mentioned below. ARS-v2 uses additional normalization for the states. We confirm from line 8 that they are calculating the mean and covariance of all states encountered by the algorithm upto that point and using it in line 5 for normalization. As an enhancement, the authors select only $b$ out of the $N$ directions.

In line 7, the update step is very similar to SFR, except the authors, in the denominator, use $\sigma_R = $ standard deviation of returns from each perturbed policy. Instead, in our implementation, we just use the standard deviation of perturbations $\nu$. The update step at line 7 then becomes the following.

$$M_{j+1} = M_j + \frac{\alpha}{b\nu} \sum_{k=1}^{b} \left[ r(\pi_{j,(k),+}) - r(\pi_{j,(k),-}) \right] \delta_{(k)}.$$

## C  Numerical Experiments: Setting Parameters

### C.1  Gridworld Details

To enable a finite horizon setup, we limit the number of steps that can be taken in the environment. A large limit results in increased runtime, so we set values as so.

| Grid size | Max episode length | Max Interactions Limit |
|:---:|:---:|:---:|
| $4 \times 4$ | 50 | 50000 |
| $10 \times 10$ | 100 | 150000 |
| $50 \times 50$ | 200 | 200000 |

Table 6: Maximum steps allowed in an episode for a given grid-size (L)

### C.1.1  Policy function

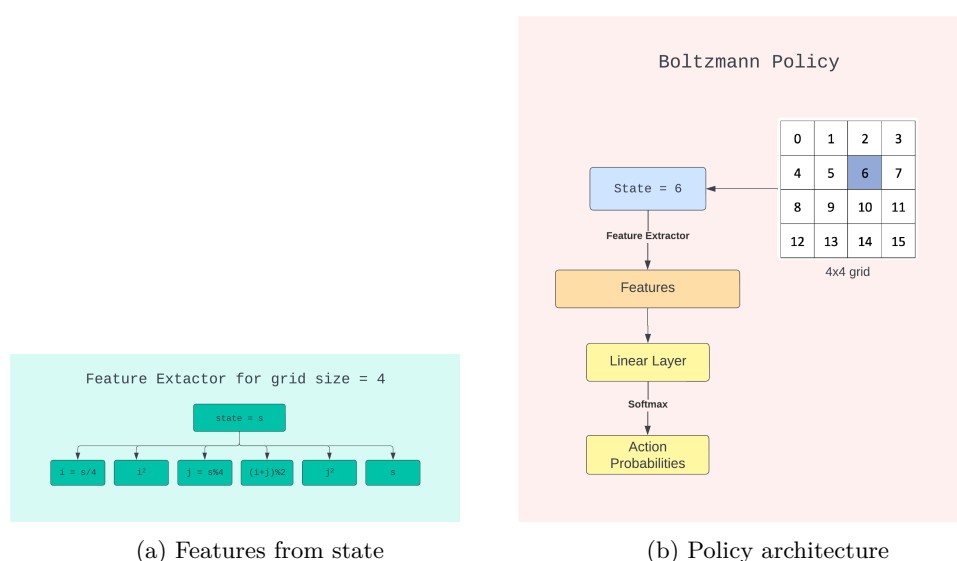

(a) Features from state  (b) Policy architecture

Figure 8:  Boltzmann Policy used across all algorithms

The input to the policy consists of polynomial and modulo features of the agent's position in the grid. Both algorithms utilize the same policy function, which computes logits for the available actions using a linear function. The logits are converted into probabilities via softmax. Figure 8 illustrates the same.

### C.1.2  Algorithm Parameters

For the SF-REINFORCE algorithm we chose the parameters $\delta(n) = \delta_0(\frac{1}{50000+n})^d$ and $\alpha(n) = \frac{\alpha_0 * 50000}{50000+n}$, where $n$ is the episode number. Here, we set $\alpha(0) = \alpha_0 = 2 \times 10^{-6}$. In this setting, $d < 0.5$ is required for convergence. We experiment with two different schemes – a decay scheme where we vary $d$ and set $\delta(0) = \delta_0 = 1$, and a constant scheme with $d = 0$ and with varying $\delta_0$. To reduce the variance, we measure $G_n$ as the average over 10 trials. The two sided version uses 5 trials for each side, allowing a fair comparison.

We also include the PPO, TRPO and A2C algorithms from Raffin et al. (2021) for our comparison. We use the same policy architecture along with a linear layer for the value function. For REINFORCE and PPO algorithms, the policy is updated using the policy gradient via an ADAM optimizer with its learning rate set to 0.0003. To ensure the results are reproducible, we run every setting 10 times with different seeds, and we plot the mean as the dark line, and shade using the standard deviation around it.

## C.2 MuJoCo Details

### C.2.1 Optimal Hyperpameters

| delta_std | deltas_used | n_directions | step_size | transform |
|-----------|-------------|--------------|-----------|-----------|
| 0.020 | 20 | 120 | 0.020 | component_clip |
| 0.020 | 20 | 120 | 0.020 | none |
| 0.020 | 20 | 40 | 0.020 | norm_clip |
| 0.020 | 20 | 80 | 0.020 | signed |

Table 7: ARS-v1 best hyperparameters for HalfCheetah

| delta_std | deltas_used | n_directions | step_size | transform |
|-----------|-------------|--------------|-----------|-----------|
| 0.020 | 20 | 80 | 0.020 | component_clip |
| 0.020 | 20 | 80 | 0.020 | none |
| 0.020 | 20 | 80 | 0.020 | norm_clip |
| 0.020 | 20 | 40 | 0.020 | signed |

Table 8: ARS-v2 best hyperparameters for HalfCheetah

| delta_std | deltas_used | n_directions | step_size | transform |
|-----------|-------------|--------------|-----------|-----------|
| 0.030 | 8 | 32 | 0.020 | component_clip |
| 0.030 | 8 | 16 | 0.020 | none |
| 0.020 | 8 | 32 | 0.020 | norm_clip |
| 0.020 | 8 | 16 | 0.020 | signed |

Table 9: ARS-v1 best hyperparameters for Hopper

| delta_std | deltas_used | n_directions | step_size | transform |
|-----------|-------------|--------------|-----------|-----------|
| 0.030 | 8 | 32 | 0.020 | component_clip |
| 0.030 | 8 | 32 | 0.020 | none |
| 0.030 | 8 | 16 | 0.020 | norm_clip |
| 0.020 | 8 | 32 | 0.020 | signed |

Table 10: ARS-v2 best hyperparameters for Hopper

| delta_std | deltas_used | n_directions | step_size | transform |
|-----------|-------------|--------------|-----------|-----------|
| 0.020 | 30 | 30 | 0.020 | component_clip |
| 0.020 | 10 | 50 | 0.020 | none |
| 0.020 | 30 | 30 | 0.020 | norm_clip |
| 0.020 | 10 | 30 | 0.020 | signed |

Table 11: ARS-v1 best hyperparameters for Swimmer

| delta_std | deltas_used | n_directions | step_size | transform |
|-----------|-------------|--------------|-----------|-----------|
| 0.020 | 10 | 30 | 0.020 | component_clip |
| 0.020 | 10 | 50 | 0.020 | none |
| 0.020 | 30 | 30 | 0.020 | norm_clip |
| 0.020 | 30 | 50 | 0.020 | signed |

Table 12: ARS-v2 best hyperparameters for Swimmer

| delta_std | deltas_used | n_directions | step_size | transform |
|-----------|-------------|--------------|-----------|-----------|
| 0.025 | 40 | 80 | 0.030 | component_clip |
| 0.025 | 40 | 80 | 0.020 | none |
| 0.025 | 40 | 100 | 0.020 | norm_clip |
| 0.025 | 40 | 80 | 0.020 | signed |

Table 13: ARS-v1 best hyperparameters for Walker2d

| delta_std | deltas_used | n_directions | step_size | transform |
|-----------|-------------|--------------|-----------|-----------|
| 0.025 | 40 | 80 | 0.030 | component_clip |
| 0.025 | 40 | 80 | 0.030 | none |
| 0.025 | 40 | 80 | 0.030 | norm_clip |
| 0.025 | 40 | 100 | 0.020 | signed |

Table 14: ARS-v2 best hyperparameters for Walker2d

| delta_std | step_size | transform |
|-----------|-----------|-----------|
| 0.020 | 0.005 | component_clip |
| 0.020 | 0.005 | no_transform |
| 0.020 | 0.010 | norm_clip |
| 0.020 | 0.040 | signed |

Table 15: SFR best hyperparameters for HalfCheetah

| delta_std | step_size | transform |
|-----------|-----------|-----------|
| 0.020 | 0.005 | component_clip |
| 0.020 | 0.005 | no_transform |
| 0.020 | 0.005 | norm_clip |
| 0.020 | 0.005 | signed |

Table 16: SFR best hyperparameters for Hopper

| delta_std | step_size | transform |
|-----------|-----------|-----------|
| 0.020 | 0.020 | component_clip |
| 0.020 | 0.020 | no_transform |
| 0.020 | 0.020 | norm_clip |
| 0.020 | 0.020 | signed |

Table 17: SFR best hyperparameters for Swimmer

| delta_std | step_size | transform |
|-----------|-----------|-----------|
| 0.025 | 0.003 | component_clip |
| 0.025 | 0.003 | no_transform |
| 0.025 | 0.003 | norm_clip |
| 0.025 | 0.003 | signed |

Table 18: SFR best hyperparameters for Walker2d

### C.2.2 All hyperparameters

| Parameter | Value |
|---|---|
| $k$ | [40, 80, 120] |
| $b$ | [40, 80, 20, 120] |
| $\alpha$ | 0.02 |
| $\nu$ | 0.02 |
| transform | [norm_clip, component_clip, signed, none] |

Table 19: ARS all hyperparameters for HalfCheetah

| Parameter | Value |
|---|---|
| $k$ | [32, 16] |
| $b$ | [8, 16, 32] |
| $\alpha$ | 0.02 |
| $\nu$ | [0.02, 0.03] |
| transform | [norm_clip, component_clip, signed, none] |

Table 20: ARS all hyperparameters for Hopper

| Parameter | Value |
|---|---|
| $k$ | [50, 30] |
| $b$ | [10, 50, 30] |
| $\alpha$ | 0.02 |
| $\nu$ | [0.02, 0.01] |
| transform | [component_clip, signed, none, norm_clip] |

Table 21: ARS all hyperparameters for Swimmer

| Parameter | Value |
|---|---|
| $k$ | [80, 100] |
| $b$ | [40, 80, 100] |
| $\alpha$ | [0.03, 0.02] |
| $\nu$ | 0.025 |
| transform | [component_clip, signed, none, norm_clip] |

Table 22: ARS all hyperparameters for Walker2d

| Parameter | Value |
|---|---|
| $\alpha$ | [0.04, 0.01, 0.005, 0.03] |
| $\nu$ | 0.02 |
| transform | [norm_clip, component_clip, signed, none] |

Table 23: SFR-2 all hyperparameters for HalfCheetah

| Parameter | Value |
|---|---|
| $\alpha$ | [0.001, 0.005, 0.01] |
| $\nu$ | [0.02, 0.025] |
| transform | [norm_clip, component_clip, signed, none] |

Table 24: SFR-2 all hyperparameters for Hopper

| Parameter | Value |
|---|---|
| $\alpha$ | 0.02 |
| $\nu$ | [0.02, 0.01] |
| transform | [norm_clip, component_clip, signed, none] |

Table 25: SFR-2 all hyperparameters for Swimmer

| Parameter | Value |
|---|---|
| $\alpha$ | [0.003, 0.01] |
| $\nu$ | [0.025, 0.02] |
| transform | [component_clip, signed, none, norm_clip] |

Table 26: SFR-2 all hyperparameters for Walker2d

