# OpenReview forum: "Variance Reduced Smoothed Functional REINFORCE Policy Gradient Algorithms"
_TMLR — Accepted by TMLR_

### Review · Reviewer_VHbd · 2025-05-27

**Summary Of Contributions:**

This paper proposes a family of smoothed functional (SF) estimators to replace REINFORCE style policy gradients for policy optimization in episodic reinforcement learning tasks. Following are the main contributions in this work.

1. Two main smoothed functional estimators are introduced:

    - SFR-1: A one-sample SF estimator.
    - SFR-2: A two-sample central-difference-style SF estimator using perturbations in opposite directions.

    These variants require only one or two state-action-cost trajectory simulations per update, making them sample-efficient.

2. To reduce the variance of parameter updates arising from noisy return estimates, SFR-1 and SFR-2 are further modified as follows:

    - Signed updates: Use only the sign of the estimated gradient components.
    - Clipped gradients: Clip the total or component wise gradient norm to control the magnitude of the update and handle outliers.

3. This paper provides asymptotic convergence proofs for all the algorithmic variants, including signed and clipped forms, as well as batched versions (ES-style).

4. Experiments are conducted on four MuJoCo continuous control tasks: Swimmer, Hopper, HalfCheetah, and Walker2d.
Given a fixed budget of environment interactions, SFR-2 (especially with component-wise clipped updates) is shown to be competitive with or superior to ARS, particularly in HalfCheetah and Swimmer.
The results suggest that frequent updates with low-variance estimators can be more effective than infrequent but high-precision updates (as in ARS).

The algorithms in this paper do not require gradient computation or critic training.
They are applicable to general episodic MDPs and nonlinear policies, making them suitable for a wide range of RL problems.

**Audience:**

Yes

**Broader Impact Concerns:**

N.A.

**Claims And Evidence:**

No

**Requested Changes:**

Following requests are critical for positive recommendation:

1. *Clarify motivation for proposed approaches*. Authors should explicitly discuss why REINFORCE is still relevant in light of more modern policy gradient methods. For instance, highlight benefits like simplicity, parallelizing ability, or test-time usage.

2. Even if the computational cost is higher, authors should include at least one run of PPO or A2C to demonstrate how proposed methods compare with widely adopted standards in continuous control.

3. Discuss applicability to other settings, for example non-episodic (discounted) MDPs or partially observable tasks.

4. Discuss possible limitations in high dimensional spaces. For example, are there certain conditions in the optimization landscape that especially disadvantage or favor SFR-2 over ARS or PG?

**Strengths And Weaknesses:**

**Strengths**:

1. This paper provides theoretical convergence proofs using stochastic approximation theory for all algorithmic variants.

2. Proposed algorithms address variance reduction using signed and clipped variants that yield empirical performance gains.

3. Empirical results compare sample efficiency over prior baselines with evaluations under a fixed budget of environment interactions in MuJoCo continuous control environments.


**Weaknesses**:

1. This paper does not quite motivate why an alternative approach to REINFORCE-style policy gradients is being studied, when there are well established algorithms like PPO, A2C etc. It would improve the quality of the paper to include some intuitive toy examples that provide insights into the differences between SF and PG methods and conditions under which one paradigm outshines the other.

2. The clarity of presentation of the results should be improved. For example, Table 2 does not highlight the best-performing algorithm in each case, making it harder to follow along with the authors' description of the results.

3. I have briefly skimmed the proofs in the Appendix. It would help improve the appeal of the techniques proposed in this paper to a wider audience if there were additional proof sketches that intuitively explain the overall flow of how the proof proceeds.
    - Possible typo in page 19: fourth line from the end of the page says "\delta_n in the denominator" - should it be "numerator" instead?


4. This paper stops at proving asymptotic convergence. Finite-time or sample complexity guarantees are left for future work, despite claiming improved sample efficiency with proposed algorithms.

---

> ### Author Response · Authors · 2025-06-20
> **Response to Reviewer VHbd**
>
> We thank the reviewer for all the comments. We have submitted a revised version of the manuscript. Below is our response to the issues raised by the reviewer.
>
> Weaknesses:
>
> We have now added a motivation for using zeroth order algorithms as alternatives to policy gradient methods in the Introduction section. Further, we have now made detailed performance comparisons with A2C, PPO, and TRPO, in addition to the ARS algorithms. We have made these performance comparisons on MuJoCo environments and also added a simpler gridworld environment (with varying grid sizes). We observe that our algorithm SFR-2 performs uniformly better than TRPO, PPO and A2C on the gridworld environment. Of the four MuJoCo environments, SFR-2 is seen to be better than TRPO, A2C and PPO on three environments, see Tables 1, 2 and 5 of the revised paper.
>
> 2. For improved clarity, we have now marked the best performing algorithm in each table in bold.
>
> 3. We have now included an outline of the proof sketch in Section 5 that also highlights the novel aspects of the analysis in relation to other work. Regarding possible typo, we have verified that the quantity $\delta_n$ in the first bias term is indeed in the denominator while in the second bias term, the same is in the numerator as can be verified through a Taylor's expansion of the value function.
>
> 4. Indeed we mention performing sample complexity analysis as future work. We also highlight now the limitations including the strong assumptions that existing finite sample analyses make on the performance objective and the system dynamics.
>
>
> Requested Changes:
>
> 1. We have now provided more motivation for the proposed approaches in the Introduction. We have also now made a subsection on `Our Contributions' in the Introduction section that highlights the significance of our algorithms. A key highlight mentioned there is on the empirical performance: ``We show empirical results on four different MuJoCo environments as well as on a gridworld environment with varying grid sizes. Our experiments demonstrate that our algorithms with the clipped and signed updates perform better than ARS on more than half of the settings. Our algorithm SFR-2 shows uniformly better performance than  A2C, PPO and TRPO on all the gridworld settings. On the MuJoCo environments, SFR-2 shows the best performance on two of the four MuJoCo environments (when ARS is also considered) and it is better than A2C, TRPO and PPO on three of these environments.''
>
> 2. Indeed we have now implemented A2C, TRPO and PPO on all the four MuJoCo environments. We have also shown implementations of these algorithms on gridworld settings with different grid sizes.
>
> 3. Our zeroth-order gradient estimation procedure would work even for continuing task problems such as those involving infinite horizon discounted reward. We have now mentioned the following in the Introduction section (see the first para on page 2): ``We may however mention that our gradient estimation procedure would work even for continuing task settings such as those involving the infinite horizon discounted reward criterion. A policy gradient procedure for this setting is described in Paternain et al. (2022), see Algorithms 1-2 there. The gradient estimator there involves the computation of a complex matrix-valued function. Zeroth-order methods can potentially help in this process as one would not need to compute such quantities."
>
> 4. One of the challenges with standard policy gradient algorithms as with the earlier proposed perturbation analysis and likelihood ratio methods for simulation optimization is that being a first order (gradient-based) method, it relies on an interchange between the gradient and expectation operators. While such an interchange is justified in the setting of finite state-action spaces, it is not so in the case of infinite state spaces. In the latter setting, one requires strong regularity conditions on the sample performance as well as system dynamics under which such an interchange between the aforementioned operators can be justified. We have now mentioned this in the Introduction section (first para on page 2). The zeroth-order algorithms such as SFR do not suffer from this problem as the expectation is directly obtained from the effects of the stochastic approximation scheme and the gradient (in the limit) is then estimated from the converged expected values, thereby invalidating the need for an interchange between the gradient and expectation operators.

---

### Review · Reviewer_Me2L · 2025-05-28

**Summary Of Contributions:**

The paper revisits the REINFORCE algorithm, proposing two variants based on single and two function measurements at a perturbed parameter value. They also propose two modifications to these algorithms for reducing their variance. The asymptotic convergence of all algorithms is shown, and extensive evaluation through simulation experiments is performed.

**Audience:**

Yes

**Claims And Evidence:**

Yes

**Requested Changes:**

Please see the weaknesses above.

**Strengths And Weaknesses:**

## Strengths

The convergence analysis of the algorithms is rigorous.

## Weaknesses

- The write-up has a number of typos, e.g., in Sec 3 *run using the policy parameter* n -> n-1.
- The structure of the paper could be improved, e.g., subsections in intro and related work section.
- Notation consistency, e.g., i Sec 3 mth trajectory at first nth trajectory a bit later, and then m is used again to denote a set as well as enumeration.
- Better presentation overall, e.g., caption inf figure is missing/bold the table where values are best
- Proposed algorithms and modifications are not motivated or intuitevely explained. E.g. why the +- in SFR-2 and not two different \Deltas? What is the reasoning behind this? Or why update rule (6) specifically?
- Empirical results are marginal. Even for the cases where SFR approaches have some gain the signal is not strong.

---

> ### Author Response · Authors · 2025-06-20
> **Response to Reviewer Me2L**
>
> We thank the reviewer for all the comments. We have submitted a revised version of the manuscript. Below is our response to the issues raised by the reviewer.
>
> Weaknesses and Requested Changes:
>
> 1. We agree with the reviewer and have now corrected the error in the time/update index as mentioned.
>
> 2. We have now added two subsections in Section 1, namely on our contributions and organization of the paper, respectively.
>
> 3. We agree there was confusion in the notation being used in Section 3 previously. We now consistently refer to the n'th trajectory in the description.
>
> 4. We have now improved on the presentation in the paper and made corrections at the mentioned points. In particular, we have added captions to the figures where the captions were earlier not present and have also identified in bold the best performing algorithm for the various experiments.
>
> 5. We have added the following line after (6) to make things clearer: ``One may view (5)-(6) as zeroth order policy gradient algorithms involving one and two measurement gradient estimators where the performance gradient is estimated using direct function measurements. "We indeed use the same $\Delta$ in both simulations in SFR-2 as it helps in bias cancellation, see the Taylor's expansions of the value functions for the two perturbed  parameter values in the proof of Proposition 2 in Appendix A.2.
> \item After we have identified the best performing algorithm in the various tables and from the various training plots, it can be seen that SFR with signed updates and clipped gradients performs better than ARS on a significant number of settings. It is also better than TRPO, PPO and A2C on various settings over which we performed our experiments. We have also now included the results of experiments on the gridworld environment with different state sizes as well as shown performance comparisons with A2C, TRPO and PPO on the MuJoCo environments, in addition to ARS. SFR-2 is seen to show consistent performance and quite often is the winner algorithm. ARS on the other hand, does not show consistent results, see for instance, Table 5.

---

### Review · Reviewer_Mkv4 · 2025-06-12

**Summary Of Contributions:**

This paper revisits the REINFORCE policy gradient algorithm and proposes a family of smoothed functional (SF) variants to reduce gradient estimator variance and improve convergence in episodic reinforcement learning tasks. The authors discuss two main estimators: a one-simulation variant (SFR-1) and a two-simulation variant (SFR-2), and augment them with signed gradient updates and clipping techniques inspired by PPO. The paper provides rigorous asymptotic convergence proofs for all proposed algorithms and empirically benchmarks them against the Augmented Random Search (ARS) baseline across several MuJoCo environments.

**Audience:**

Yes

**Broader Impact Concerns:**

None.

**Claims And Evidence:**

Yes

**Requested Changes:**

Suggestions for Improvement:

- Clarify contributions: Clearly differentiate what is new (e.g., convergence analysis, signed/clipped variants in an RL context) versus what is a known technique (e.g., SF gradient estimators). It seems that all the algorithms considered are well known and well studied. Please clarify the impact of the theoretical and empirical results.

- The tables are dense. To make them easier to read, I recommend bolding the highest performing numbers to make it easier to see trends in which algorithms work best. In particular, it is hard to tell which method is the best or in which cases an algorithm should be recommended.

- Improve accessibility by possibly adding a diagram summarizing the algorithm family, include intuitive explanations before/after formal results to emphasize why the theory is valuable/interesting, and simplify notation where possible.

- A summary of the convergence results at the end of section 5 would be very beneficial. In particular, what would we expect in terms of performance based on the theory? Does the theory match what is seen in the empirical results of section 6? What is learned from section 5 that was previously unknown?

- Plot cumulative rewards vs. environment interactions to visualize convergence dynamics and variance across seeds.

- Minor comment, but I found the citations to be awkward and I think they should be fixed and made consistent. On the second page parenthetical citations are used as nouns:
For example: "In (Salimans et al., 2017), evolutionary..." I would recommend something like "In Salimans et al. (2017), evolutionary..." which you should be able to get with \citet.
In the first couple paragraphs the opposite happens where citations that are usually parentheticals are used without parentheses.
For example: "The policy gradient theorem, cf. Sutton et al. (1999); Marbach & Tsitsiklis (2001); Cao (2007),
which is a fundamental result..."
I would recommend just putting these as \citep. For example: "The policy gradient theorem (Sutton et al. 1999; Marbach & Tsitsiklis 2001; Cao 2007), which is a fundamental result..."
I think this will make the citations more standard.

- The figures above Table 1 should have a caption and be referenced in the text.

**Strengths And Weaknesses:**

Strengths:

+ The paper offers asymptotic convergence analysis for the proposed algorithms.

+ Incorporating signed gradients and clipping into the SF-REINFORCE framework is shown to leads to improvements in stability and sample efficiency. While this is probably to be expected, it is nice to demonstrate empirically.

+ The paper compares their approach with ARS on multiple MuJoCo benchmarks. The results suggest that the proposed variants, particularly SFR-2 with clipping or signed updates, are competitive or even superior in certain environments.

+ Good motivation and connection to existing literature on random search, policy gradient, and zero-order optimization.

Weaknesses:

- The paper is mathematically dense, especially in the convergence analysis sections, which may hinder accessibility for broader RL audiences. Furthermore, it uses notation that is non-standard compared to most machine learning papers on RL.

- A lot of the formalization in section 2 seems overly dense and unnecessary. The paper assumes "that the policies are stationary randomized and are parameterized via a certain parameter θ" so I think things would be more succinct and accessible if a lot of the earlier notation was dropped and standardized. For example, why use $\phi_\theta(s, a)$ when $\pi(a|s)$ is much more common in the literature.

- The paper primarily reports final return values in tables; learning curves or training dynamics over time or other visuals would provide more insight into stability and convergence behavior.

- It's unclear what the main contributions are and what recommendations should be taken from the paper in terms of which algorithms to use and when. From the paper it seems the theory is the main stated contribution, but what about the theory is important? Is it just the fact that the paper shows asymptotic convergence? Was this believed to not be true? What are the main benefits of having this new theory?

---

> ### Author Response · Authors · 2025-06-20
> **Response to Reviewer Mkv4**
>
> We thank the reviewer for all the comments. We have submitted a revised version of the manuscript. Below is our response to the issues raised by the reviewer.
>
> Weaknesses:
>
> 1.We now provide a high-level overview of the proof structure at the beginning of Section 5.
>
> 2. We now mention both in the introduction as well as the proof sketch given in Section 5 that ``We prove the asymptotic convergence of SFR-1 and SFR-2 under just two assumptions, namely Assumptions 1 and 2. In fact, we prove all the basic requirements such as the parameterized value function being differentiable with a Lipschitz continuous gradient (Lemma 4). This is unlike papers on ES/ARS (Malik et al (2020)) that make much stronger assumptions but do not prove whether such assumptions are valid in the settings they consider.
>
> 3. The formalism in Section 2 is indeed required. We also provide an important assumption that is used for stochastic shortest path problems, namely, that all policies are proper. We do not use $\pi(a|s)$ for a randomized policy because we use the notation \pi for a deterministic policy. We believe using the same notation for both deterministic and randomized policies will cause confusion.
>
> 4. We have now also included training/learning curves plotted after accounting for runs with different initial seeds to depict the stability of the algorithms. Further, we have shown performance comparisons with PPO, TRPO and A2C algorithms (using both learning curves as well as tabular comparisons), in addition to the ARS algorithms.
>
> 5.  We highlight the main benefits of applying SFR-2. (i) SFR-2 performs better than A2C, TRPO and PPO on majority of the settings. (ii) Variants of SFR-2 based on clipping and signed updates perform better than the ARS algorithms on a majority of settings. This is despite the fact that ARS requires $2k$ different simulation runs as opposed to just 2. Thus, for the same number of parameter updates, ARS requires far more computation. This we believe is a major advantage with our algorithm. Regarding the theoretical analysis, note that it's not just the asymptotic convergence that we show but also several smaller results that are truly significant and important. To begin with we prove our regularity requirements such as the function V(.) being differentiable and the gradient being Lispchitz continuous. Such results are not present in the literature. Other papers such as Malik et al (2020), provide a finite time sample complexity analysis but under much stronger assumptions. In fact, all currently available sample complexity analyses make such assumptions. We also summarize the key contributions in the revised paper in Section 1.1.
>
> Requested Changes:
>
> 1. We have added a subsection in Section 1 on our contributions where we highlight the significance of both our theoretical and empirical results.
>
> 2.  We have now marked the best performing values obtained in the various tables in bold to highlight the best performing algorithm for each experiment.
>
> 3.  We have given now a broad overview of the proof structure and significance of the results we obtain at the beginning of Section 5 (Convergence Analysis).
>
> 4. The analysis in Section 5 presents many new/novel aspects. For instance, (i) we prove in Lemma 4 that the parameterized value function is continuously differentiable with its gradient being Lipschitz continuous. We also show that our algorithms involving signed updates and clipped gradient estimators have lower variance (see Lemmas 1 and 2). Moreover, our analysis of the signed version of the algorithm is new to the literature. We also prove the asymptotic convergence of the ES/ARS algorithms that had previously not been shown. What was available previously was the finite sample analysis of these algorithms, that too under majorly restrictive assumptions. We have discussed this point in more detail in the paper. Our results show that the algorithms converge asymptotically to local maxima. Note that unlike other papers, we do not make any assumptions such as on the convexity of the objective function and constraints or system dynamics and cost structure such as linear systems with quadratic costs. Our setting involves fully nonlinear functions that are not known a priori as these are expectations over noisy samples of the performance objective. Thus, in our setting, one will typically encounter multiple local maxima, minima and saddle points. This is a far more difficult setting to analyze than what the related papers consider.
>
> 5. We have now provided the training plots of the various algorithms.
>
> 6. We have now corrected the paper citation style or format at various places in the paper.
>
> 7. We have now added captions for figures above Table 1 and also the new figures for the experiments run for comparisons on both MuJoCo and the new gridworld environment. These new experimental results have been included in the revised paper.

---

### Decision · Action_Editor_dFda · 2025-07-28

**Recommendation:** Accept as is

**Audience:**

Yes

**Audience Explanation:**

This paper can be of interest to a general RL audience, in particular those in theoretical RL. The audience was broadened after the authors addressed reviewer feedback and provided more high-level insights.

**Claims And Evidence:**

Yes

**Claims Explanation:**

This paper presents a thorough theoretical analysis of variants of policy gradient methods, aimed at reducing gradient variance and improve convergence. Based on reviewer feedback, the authors have clarified the theoretical contributions to make them more widely accessible.

---

> ### Author Response · Authors · 2025-07-31
> **Thanks!**
>
> We wish to thank the editor and all reviewers for your many comments that have helped improve our work in various ways. Thank you all!